# Long-term response of oceanic carbon uptake to global warming via physical and biological pumps

Akitomo Yamamoto[1,2], Ayako Abe-Ouchi[1,2], Yasuhiro Yamanaka[3]

[1] Atmospheric an Ocean Research Institute, University of Tokyo, Kashiwa, Japan
[2]Japan Agency for Marine-Earth Science and Technology, Yokohama, Japan
[3]Graduate School of Environmental Science, Hokkaido University, Sapporo, Japan

*Correspondence to*: A. Yamamoto (akitomo@jamstec.go.jp)

**Abstract.** Global warming is expected to significantly decrease oceanic carbon uptake and therefore increase atmospheric $CO_2$ and global warming. The primary reasons given in previous studies for such changes in the oceanic carbon uptake are the solubility reduction due to seawater warming and changes in the ocean circulation and biological pump. However, the quantitative contributions of different processes to the overall reduction in ocean uptake are still unclear. In this study, we investigated multi-millennium responses of oceanic carbon uptake to global warming and quantified the contributions of the physical and biological pumps to these responses using an atmosphere-ocean general circulation model and a biogeochemical model. We found that global warming reduced oceanic $CO_2$ uptake by 13% (30%) in the first 140 years (after 2000 model years), consistent with previous studies. Our sensitivity experiments showed that this reduction is primarily driven by changes in the organic matter cycle via ocean-circulation change and solubility change due to seawater warming. These results differ from most previous studies, in which circulation changes and solubility change from seawater warming are the dominant processes. The weakening of biological production and carbon export induced by circulation change and lower nutrient supply, diminishes the vertical DIC gradient and substantially reduces the $CO_2$ uptake. The weaker deep-ocean circulation decreases the downward transport of $CO_2$ from the surface to the deep ocean, leading to a drop in $CO_2$ uptake in high-latitude regions. Conversely, weaker equatorial upwelling reduces the upward transport of natural $CO_2$ and therefore enhances the $CO_2$ uptake in low-latitude regions. Because these effects cancel each other, circulation change plays only a small direct role in the reduction of $CO_2$ uptake due to global warming, but a large indirect role through nutrient transport and biological processes.

## 1 Introduction

Since the beginning of the industrial era, global oceans have reduced the atmospheric $CO_2$ concentration and mitigated climate change by taking up approximately 25% of anthropogenic $CO_2$ emissions (Ciais et al., 2014). In contrast to terrestrial uptake, oceanic carbon uptake operates on millennial time scales (Plattner et al., 2008; Archer et al., 2009a; Zickfeld et al., 2013). Therefore, understanding changes in the ocean carbon cycle is crucial for predicting the long-term evolution of climate system components with a slow response time, such as ice sheets (Charbit et al., 2008), permafrost (Lawrence and Slater, 2005), and methane hydrate in continental margins (Archer et al., 2009b; Yamamoto et al., 2014) as well as atmospheric $CO_2$ concentration.

Climate warming tends to reduce the oceanic uptake of $CO_2$ from the atmosphere, thereby increasing $CO_2$ accumulation in the atmosphere and causing further warming, a well-known positive climate-carbon cycle feedback (Cox et al., 2000; Dufresne et al., 2002; Friedlingstein et al., 2006; Randerson et al., 2015). Changes in $CO_2$ solubility due to seawater warming, ocean circulation, and the biological pump alter oceanic $CO_2$ uptake by affecting anthropogenic $CO_2$ uptake and altering the air-sea balance of the natural $CO_2$ cycle. Most previous studies have shown that solubility change from seawater warming and circulation change are major contributors to the reduction in oceanic carbon uptake (Sarmiento et al., 1998; Joos et al., 1999; Matear and Hirst, 1999; Plattner et al., 2001; Matsumoto et al., 2010). In such previous studies, changes in the biological pump associated with ocean circulation change were regarded as a second-order process even though the biological pump plays a crucial role in the natural carbon cycle.

On the other hand, changes in the biological pump can potentially contribute significantly to a reduction in oceanic carbon uptake, considering that both previous and current generations of Earth system models (ESMs) show a consistent decrease in global mean export production between −6% and −20% by the year 2100 under strong climate warming (Steinacher et al., 2010; Bopp et al., 2013). In the natural carbon cycle, the biological pump maintains the surface-to-deep gradient in oceanic DIC, and therefore regulates the exchange of $CO_2$ between the atmosphere and the ocean. Model simulations suggested that a complete die-off of ocean life would lead to an atmospheric $CO_2$ increase of more than 150 ppm (Maier-Reimer et al., 1996; Sarmiento and Gruber, 2006). A significant reduction in $CO_2$ uptake associated with the weakening of the biological pump was found in Zickfeld et al. (2008) even though their investigation only focused on the uptake change due to the weakening of the Atlantic meridional overturning circulation (AMOC) by freshwater input.

To our knowledge, the contributions of both physical and biological effects to reductions in oceanic $CO_2$ uptake are not evaluated directly in the recent generation of coupled atmosphere-ocean general circulation models (AOGCMs) and ocean biogeochemical models. Moreover, previous studies using Earth System Models of Intermediate Complexity (EMICs) show that global warming decreases oceanic $CO_2$ uptake continuously over a thousand years or more (Plattner et al., 2008;

Schmittner et al., 2008; Archer et al., 2009a). These millennial-scale simulations are now feasible with an AOGCM due to increased computer power. Thus we examined multi-millennium changes in oceanic carbon uptake due to global warming and the contribution of individual mechanisms to these uptake change using a series of global warming simulations conducted with an AOGCM and an offline ocean biogeochemical model. The methods applied here enable us to conduct multiple simulations of the ocean carbon cycle according to AOGCM climate simulations with low computational cost.

An important difference between most previous studies (e.g., Sarmiento et al., 1998; Joos et al., 1999; Matear and Hirst, 1999; Plattner et al., 2001; Matsumoto et al., 2010) and that of Zickfeld et al. (2008) is the experimental setup used to quantify the contribution of individual mechanisms to changes in oceanic carbon uptake. In the typical former method (Sarmient et al., 1998; Matsumoto et al., 2010), abiotic experiments (i.e., without a biological pump) were conducted to isolate physical processes from biological processes. Zickfeld et al. (2008) separated these two effects more directly. In their sensitivity studies, the archived source and sink terms for biological processes in pre-industrial climate experiments were replaced with those of global warming experiments. In our study, in order to assess the different feedback mechanisms arising from different experimental setups, we quantified the contributions of individual mechanisms to changes in oceanic carbon uptake based on both abiotic experiments and methods of Zickfeld et al. (2008). We then compared these results to discuss which experimental setup is more useful for quantification of feedback mechanisms.

The remainder of this paper is organized as follows. Section 2 introduces the models and experiment design. Section 3 describes the centennial- and millennial-scale changes in the climate and ocean carbon cycle. Section 4 presents the results of three sets of sensitivity studies that quantify the contributions of individual mechanisms to reductions in oceanic $CO_2$ uptake. Here we also discuss causes for differing results arising from the different experimental setups before summarizing our research in Section 5.

## 2 Methods

### 2.1 Climate model

We used the global-warming experiments conducted by Yamamoto et al (2015) with the MIROC 4m AOGCM. This model is based on the MIROC 3.2 (K-1 model developers, 2004) that contributed to the Coupled Model Intercomparison Project phase 3 (Meehl et al., 2007), which was extensively cited in the Intergovernmental Panel on Climate Change Fourth Assessment Report. In the MIROC 4m, the coefficient of the isopycnal layer thickness diffusivity is set as $7.0 \times 10^{-6}$ cm$^2$ s$^{-1}$ instead of the value of $3.0 \times 10^{-6}$ cm$^2$ s$^{-1}$ used in the original MIROC 3.2. The atmospheric GCM has a horizontal resolution of T42 (~2.8°) and 20 vertical levels. The oceanic GCM has a zonal resolution of 1.4° and a spatially varying meridional resolution that is approximately 0.56° at latitudes lower than 8° and 1.4° at latitudes higher than 65°; the resolution changes smoothly between these latitudes. The vertical coordinate is a hybrid of sigma-z, resolving 44 levels in total: 8 sigma-layers

near the surface, 35 z-layers beneath, and one bottom boundary layer (Nakano and Suginohara, 2002). The sea-ice component is based on a two-category thickness representation, zero-layer thermodynamics (Semtner, 1976), and dynamics using elastic-viscous-plastic rheology (Hunke and Dukowicz, 1997). A model spin-up has been performed under pre-industrial boundary conditions such as insolation and an atmospheric $CO_2$ concentration of 285 ppm.

## 5  2.2 Offline ocean biogeochemical model

The ocean carbon cycle was calculated by the MIROC-based offline biogeochemical model (Yamamoto et al., 2015) that enables us to conduct numerous sensitivity experiments of the ocean carbon cycle forced by outputs from AOGCM climate simulations with low computational cost. The horizontal and vertical resolutions are the same as in the MIROC 3.2. For the tracer calculation, the monthly output data of velocity, temperature, salinity, sea-surface height, sea-surface wind speed, sea-
ice fraction, and sea-surface solar radiation from MIROC are applied to an offline tracer scheme (Oka et al., 2011).

This biogeochemical model is a modified version of the nutrient–phytoplankton–zooplankton–detritus (NPZD) ecosystem model of Keller et al. (2012). The model includes two phytoplankton classes (nitrogen fixers and other phytoplankton), zooplankton, particulate detritus, nitrate ($NO_3$), phosphate ($PO_4$), dissolved oxygen ($O_2$), dissolved inorganic carbon (DIC),
alkalinity (ALK), two carbon isotopes ($^{13}C$ and $^{14}C$), and an ideal age tracer. Constant stoichiometry relates the C, N, and P content of the biological variables and their exchanges to the inorganic variables ($NO_3$, $PO_4$, $O_2$, ALK, and DIC). Optimal uptake kinetics, which assume a physiological trade-off between the efficiency of nutrient encountered at the cell surface and the maximum assimilation rate (Smith et al., 2009), are adopted for the nutrient limitation of the growth rate of phytoplankton. The maximum phytoplankton growth and microbial remineralization rates are assumed to increase with
seawater temperature according to Eppley (1972).

The formulations of air-sea gas exchange and carbon chemistry follow the protocols of the Ocean Carbon Cycle Model Intercomparison Project–Phase 2 (OCMIP2; http://www.ipsl.jussieu.fr/OCMIP/phase2/) (Najjar and Orr, 1999). The production rate of calcium carbonate ($CaCO_3$) is assumed to be proportional to that of particulate organic carbon (POC,
$CaCO_3$:POC = 0.03) and the settling flux of particulate $CaCO_3$ in the water column decreases with an e-folding depth of 3500 m (Schmittner et al., 2008). Although iron cycling is not explicitly calculated in our model, we adopted the iron limitation for the phytoplankton growth rate using monthly dissolved iron concentration outputs from pre-industrial experiments conducted using the biogeochemical elemental cycling model (AOU_06 case in Misumi et al., (2013)). Thus, we cannot consider changes in iron cycling and their impact on ecosystems in the global-warming experiment. This model
does not include a marine-sediment module because the reaction of the dissolved $CO_2$ with $CaCO_3$ in deep-ocean sediments only plays an important role at timescales of thousands of years or longer (Archer et al., 1998).

As we forced the offline ocean biogeochemical model with monthly mean data from the AOGCM, a certain amount of short-term variability was averaged over. To confirm the validity of the offline calculations and their sensitivity to the forcing frequency, we compared the passive salinity distribution in the offline model to online salinity distribution in the AOGCM. There were no significant differences in basin-scale distributions between the two simulations, suggesting that short-term processes had limited impact on our results.

The biogeochemical model was initialized from the annual mean climatology data based on the World Ocean Atlas 2005 (WOA2005: Garcia et al., 2006) for dissolved $NO_3$ and $PO_4$ and Global Ocean Data Analysis Project (GLODAP: Key et al., 2004) for DIC and ALK. For the spin-up, the last 50 years of data in the MIROC pre-industrial experiment were cyclically applied to the offline ocean biogeochemical model. A model spin-up of more than 8000 years has been performed in order to eliminate the model drift in the global inventory of all tracers. In the last century of the spin-up run, the global net atmosphere-ocean $CO_2$ exchange is $3.0 \times 10^{-4}$ PgC year$^{-1}$, which is smaller than the OCMIP2 threshold of 0.01 PgC year$^{-1}$ (Orr, 2002). The resultant initial storage of ocean carbon (35,400 PgC), ocean net primary production (49.6 PgC year$^{-1}$) and export production (8.1 PgC year$^{-1}$) are comparable to observations and other estimates (Antoine et al., 1996; Schlitzer, 2000; Ciais et al., 2014). The simulated basin-scale distributions of nutrients ($PO_4$ and $NO_3$), DIC, and ALK under pre-industrial conditions are generally in agreement with the modern distributions in the WOA2005 and GLODAP datasets (Fig. S1). All physical and biogeochemical tracers, except salinity, have correlation coefficients with observational data of more than 0.85 and normalized standard deviation values between 0.7 and 1.1 (Fig. S2).

## 2.3 Global warming experiment

After the spin-up, the prescribed atmospheric $CO_2$ concentration for the climate model was increased by 1% per year from the pre-industrial value (285 ppm) until it reached $4 \times CO_2$ levels (1140 ppm) at year 140. This value was then held constant until reaching 2000 model years. Using the output of the global warming experiment conducted by the climate model, the offline biogeochemical model was integrated for 2000 years (GW_bio in Table 1).

The prescribed $CO_2$ concentration during the first 140 years of our experimental design are higher than the business-as-usual scenarios such as SRES A2 and RCP8.5, so the response of climate and oceanic carbon cycle would be somewhat different. After year 140, the influence of the initial difference would weaken because of the similar $CO_2$ concentrations between the 4 $\times CO_2$ and business-as-usual scenarios. Therefore, the $CO_2$ differences between these scenarios would have a limited impact on the long-term response of climate and ocean carbon cycle to global warming. We compare our results with previous long-term simulations under the business-as-usual scenario in Section 3.

In the real world, the climate and carbon cycle change in response to $CO_2$ emissions; our simulation with prescribed $CO_2$ concentrations is thus highly idealized. However, there is an advantage to simulations with prescribed $CO_2$ concentrations as

opposed to those with prescribed emissions: the former allow for a more rigorous separation of feedback processes since carbon sinks respond to the same atmospheric $CO_2$ concentration in all simulations (Zickfeld et al., 2011).

## 2.4 Sensitivity experiments

We conducted additional experiments for the first 500 years using the offline biogeochemical model (Table 1). In the constant-climate run (CTL_bio), the same prescribed $CO_2$ concentration as used in GW_bio but with the climate conditions of the MIROC pre-industrial experiment were applied to the offline biogeochemical model. This experiment relied on the same concept as the uncopled simulation in Friedlingstein et al. (2006), in which changing atmospheric $CO_2$ concentrations did not affect the radiation balance and therefore the climate. The total change in oceanic $CO_2$ uptake due to global warming is obtained from the difference between GW_bio and CTL_bio ("Total" in Table 2). We conducted three sets of experiments to isolate the sensitivity of $CO_2$ uptake with respect to changes in sea surface temperature (SST), ocean circulation, biology, freshwater flux, and sea ice (Table 1).

### 2.4.1 Abiotic experiments

The first set (3–5 in Table 1 and "abiotic" in Table 2) are abiotic experiments, a typical method used in previous studies (Sarmient et al., 1998; Matsumoto et al., 2010). Following established methodology, the offline ocean biogeochemical model, which does not include a biological pump (no production and remineralization of POC and $CaCO_3$), is used in this first set. The abiotic offline biogeochemical model was spun up under the climate conditions of the MIROC pre-industrial experiment, as was the original offline biogeochemical model. After the spin-up, the surface-to-deep gradient of DIC decreased significantly (Fig. 1), as also shown in Sarmiento and Gruber (2006). The influence of the reduced DIC concentration in the ocean interior is discussed in Section 4.3.

CTL_abio and GW_abio are identical to CTL_bio and GW_bio, respectively, except that they are calculated using the abiotic model. Experiment GW_abio_SST is the same as GW_abio, except that the solubility is calculated using the pre-industrial SST. The SST effect is calculated from GW_abio minus GW_abio_SST, and the circulation effect is calculated from GW_abio_SST minus CTL_abio ("SST" and "Circulation", respectively, in Table 2). The biological effect is calculated as the residual ("Total" − "SST" − "Circulation", as described in Table 2). In this set, the effects of changes in freshwater flux and sea ice coverage on changes in the $CO_2$ uptake are neglected, as in previous studies.

### 2.4.2 GW-base and CTL-base experiments

The second set of sensitivity experiments (GW-base) are based on Zickfeld et al. (2008). The experiments are based on GW_bio; however, each mechanism is replaced from GW_bio to CTL_bio (6–12 in Table 1). The contributions of individual mechanisms to the $CO_2$ uptake change are calculated from the difference between GW_bio and each experiment ("GW-base"

in Table 2). Experiment GW_bio_SST is identical to GW_bio, except that pre-industrial SST is used to calculate the solubility. This experiment evaluates the influence of the solubility changes due to the SST increases on the $CO_2$ uptake.

Experiment GW_bio_circ is based on GW_bio; however, it uses the preindustrial ocean's vertical and horizontal velocities, sea surface height, and vertical diffusivity. Note that the archived monthly mean data of the source and sink of DIC and alkalinity due to the biological pump (the organic matter cycle and the $CaCO_3$ cycle) in GW_bio are applied to the offline biogeochemical model. The $CO_2$ uptake sensitivity to ocean circulation change is isolated in this experiment.

Experiments GW_bio_om and GW_bio_ca are similar to GW_bio, except that the archived monthly mean data of the source and sink of DIC and alkalinity due to the organic matter and $CaCO_3$ cycles in CTL_bio, respectively, are applied to the offline biogeochemical model. These two sensitivity studies have the same ocean circulation as GW_bio. Comparing GW_bio to the GW_bio_om ("Organic matter cycle" of GW-base in Table 2), reduced export production decreases carbon uptake but reduced remineralization decreases upward transport of remineralized carbon, leading to enhanced carbon uptake. These experiments evaluate the contributions of changes in the organic matter and $CaCO_3$ cycles. The biological effect is obtained from the sum of these two effects. Note that changes in productivity due to stratification and circulation changes are included in these biological sensitivity experiments rather than in circulation sensitivity experiments.

GW_bio_si is similar to GW_bio, except that the preindustrial sea ice fractions in both hemispheres are used. Note that the pre-industrial sea ice fractions are used only in the gas exchange calculations. The effects of biological and circulation changes due to the sea ice fractions are not included in this experiment.

GW_bio_fw is identical to GW_bio, except that the freshwater flux of the pre-industrial experiment is applied to the offline biogeochemical model. This experiment evaluates the influence of changes in salinity, DIC, and alkalinity due to the hydrological cycle change on the $CO_2$ uptake.

The contributions of individual mechanisms to the $CO_2$ uptake change are also estimated from sensitivity experiments under pre-industrial climate conditions. In the third set (CTL-base), following GW-base, sensitivity experiments are based on CTL_bio; however, each mechanism is replaced from CTL_bio to GW_bio (13–19 in Table 1). The contributions of individual mechanisms are obtained from the differences between each experiment and CTL_bio ("CTL-base" in Table 2). The differences in the estimated contributions between GW-base and CTL-base reflect the non-linearity of each mechanism to the $CO_2$ uptake change.

## 3 Multi-millennium responses to global warming

### 3.1 Climate and ocean biogeochemical cycles

In this section, we briefly describe the regional changes in climate and ocean biogeochemical variables; a full summary of global mean changes was reported in Yamamoto et al. (2015). At the end of the global warming simulation, the global mean surface air and ocean temperature increase by 8.4 °C and 6.5 °C, respectively. The AMOC decreases from 16.5 Sv to 3.8 Sv in the first 500 years and does not recover by the end of the simulation. The strength of the Antarctic Bottom Water (AABW) formation decreases from 6.2 Sv to 2.4 Sv in the first 500 years. Thereafter, the strength of the AABW formation recovers and overshoots the pre-industrial condition after year 1000. These global mean changes are comparable to previous long-term simulations with EMIC and AOGCM (see Yamamoto et al. (2015) for a more detailed discussion).

At the time of $CO_2$ quadrupling (year 140), polar amplification occurs only in the Northern Hemisphere (Fig. 2a). The surface temperature over the Southern Ocean and Antarctica gradually increases, and polar amplification in both hemispheres is observed at approximately year 1000. This slow southern polar warming is related to deep-ocean heat uptake in the Southern Ocean. According to the slow deep-ocean heat uptake, the ocean interior is still warming up at the end of the simulations (Fig. 2b). Slow southern polar warming and ocean heat uptake have also been reported in previous multi-millennial simulations conducted by ECHAM5/MPIOM (Li et al., 2013). In the ocean, greater warming of the upper ocean (above 2000 m) occurs in the Southern Hemisphere (Fig. 2c). This is because the drastically weakened AMOC and enhanced AABW formations cause the heat transport to be larger in the Southern Hemisphere than in the Northern Hemisphere (Fig. 2d). Weakening in AMOC reduces the heat transport from 1.8 PW to 0.8 PW across 30° N. By contrast, the enhanced AABW increases heat transport by 0.7 PW in the Southern Hemisphere.

Strengthening and poleward shifting of the subpolar westerlies occurs in the Southern Hemisphere (Fig. 2e), which is consistent with the observations and a robust feature of climate projections in global-warming experiments (Fyfe and Saenko, 2006; Meijers, 2014). A 12% strengthening and a 2.8° poleward shift of the maximum annual mean zonal wind stress occur at the time of $CO_2$ quadrupling. Subsequently, the subpolar westerlies in the Southern Hemisphere are gradually weakened but are still slightly stronger than the pre-industrial conditions. This weakening occurs because the slow southern polar warming weakens the thermal gradient between the tropical and polar areas. The contributions of ocean-circulation change caused by strengthened westerlies to the ocean carbon cycle are discussed in Section 4.2.

Global $PO_4$ and $NO_3$ concentration at the surface decreases by about 25% and 20%, respectively. These decreases are attributed to reduced nutrient supply into the euphotic zone due to enhanced stratification and slower deep-ocean circulation. As a result, the global export production decreases by 22% (see Fig. 1d in Yamamoto et al., 2015), which is comparable with CMIP5 models (7%–18% reduction in export production from the 1990s to 2090s for RCP8.5) (Bopp et al., 2013).

Remarkable reductions in surface PO4 concentration occurs in the tropical ocean and the North Atlantic (Fig. 2f). Consistent with this reduction, the largest declines in export production are also located around the equator and in the North Atlantic (Fig. 2g). The increases in export production are primarily observed in the Southern and Arctic Oceans. These increases are related to reductions in light limitation and/or increased growth rates due to rising temperatures as shown by Steinacher et al. (2010).

The efficiency of the oceanic biological pump is calculated following Ito and Follows (2005) as the biologically sequestered fraction of the total phosphate inventory, $P_{remi}/P_{tot}$. Here, $P_{tot}$ is the total phosphate inventory and $P_{remi}$ is remineralized phosphate concentration ($P_{remi}$=AOU $\times$ $R_{P:O}$), where AOU is the apparent oxygen utilization (AOU = $O_2^{sat}$ – $O_2$), $O_2$ is the dissolved oxygen concentration, and $R_{P:O}$ is a constant phosphorous to oxygen ratio. $P_{remi}/P_{tot}$ increases from its pre-industrial value of 0.47 to 0.51 in the first 200 years. This increased efficiency of the biological pump is consistent with CMIP5 models (calculated in terms of AOU) (Schwinger et al., 2014). After 500 model years, $P_{remi}/P_{tot}$ decreases to 0.34 by the end of the simulation. Preformed PO4 is accumulated in AABW (Fig. S3). This reduced efficiency of the biological pump is associated with enhanced AABW formation after 500 model years (Yamamoto et al., 2015).

Global primary production increases by 2.5 PgC year$^{-1}$ from the preindustrial value, which is consistent with previous studies (Schmittner et al., 2008; Taucher and Oschlies, 2011). Both diazotrophs and other phytoplankton increase slightly. Taucher and Oschlies (2011) showed that increases in primary production are caused by temperature effects on biological processes such as remineralization and the microbial loop.

Due to the constant production ratio of POC and CaCO3 in the model formulation and the increase in the global primary production, CaCO3 production is increased by 6%. The export of CaCO3 also increases in spite of a reduction in the POC export, resulting in the rain ratio increasing from 0.09 to 0.13. Two distinct export responses are caused by faster and shallower remineralization of POC in a warmer ocean; however, the remineralization rate of CaCO3 is independent of the seawater temperature in our model. The enhanced CaCO3 production decreases (increases) alkalinity in the surface (deep) oceans as shown by changes in the salinity-normalized alkalinity (dashed lines in Fig. 2h). Reduced surface alkalinity is also caused by the longer residence time of surface waters, allowing a more efficient biological utilization of the carbonate ions. The enhanced surface-to-deep gradient of the salinity-normalized alkalinity and the alkalinity associated with changes in CaCO3 cycling and ocean circulation has been reported in previous long-term simulations (Plattner et al., 2001; Schmittner et al., 2008).

Changes in the freshwater flux also alter the alkalinity. At low latitudes, enhanced evaporation concentrates the surface alkalinity. Dilution of the surface alkalinity by enhanced precipitation at high latitudes reduces the alkalinity in the deep ocean. As a result, changes in the freshwater flux causes weaker surface-to-deep alkalinity gradients. The effects of changes

in the CaCO₃ cycling and freshwater flux cancel each other out, resulting in apparently slight changes in the surface-to-deep alkalinity gradient in our model (solid lines in Fig. 2h). The effects of changes in the CaCO₃ cycle and freshwater flux to the $CO_2$ uptake change are quantitatively discussed in Section 4.1.

## 3.2 Ocean carbon cycle

We first show the cumulative oceanic uptake of $CO_2$ and the reduction of oceanic $CO_2$ uptake due to global warming. At the time of $CO_2$ quadrupling (year 140), the cumulative $CO_2$ uptake is 567 PgC in the global warming run (GW_bio) and 654 PgC in the constant climate run (CTL_bio), a projected reduction due to global warming of 13%. These values are close to those of corresponding simulations in the Coupled Model Intercomparison Project 5 (CMIP5, (Taylor et al., 2012)) in which the modeled mean $CO_2$ uptake is 613 PgC and the uptake reduction is 11% (Arora et al., 2013). Even though oceanic $CO_2$ uptake is decreased during constant atmospheric $CO_2$ after year 140, a new equilibrium is not yet reached by the end of the simulation. In the last decade of the experiments, the ocean still takes up approximately 0.31 PgC year$^{-1}$ due to the millennial timescale of the deep ocean ventilation (Key et al., 2004). At the end of the simulation, the cumulative $CO_2$ uptake is 2028 PgC and the uptake reduction due to global warming increases to 30% (Fig. 3a and Table 1). These values and the increasing trend of the uptake reduction are also comparable to the results of previous long-term EMIC simulations (Plattner et al., 2001; Schmittner et al., 2008).

The time evolutions of oceanic $CO_2$ uptake in each basin are shown in Figure 3b. During the first few decades, the Pacific Ocean is the dominant sink of atmospheric $CO_2$ because it possesses the largest area. After atmospheric $CO_2$ stops increasing, the surface Pacific waters are close to saturation and $CO_2$ uptake rapidly decreases because the deep Pacific waters are slowly ventilated. Conversely, the decrease in $CO_2$ uptake in the Southern Oceans is slower than that in the Pacific Ocean. Carbon transport from the surface to deep ocean via these well-ventilated waters causes continuous $CO_2$ uptake there. In the pre-industrial condition, the pCO₂ in the Southern Ocean is kept higher than the atmospheric pCO₂ owing to the upwelling of carbon-rich deep waters, resulting in a source of atmospheric $CO_2$. The Southern Ocean alters from a source to a sink of atmospheric $CO_2$ immediately after the atmospheric $CO_2$ increase. This phenomenon is consistent with observation-based reconstructions of present-day and pre-industrial air-sea $CO_2$ fluxes (Gruber et al., 2009). After excess $CO_2$ is mixed throughout the deep ocean, the pCO₂ in the Southern Ocean again exceeds the atmospheric pCO₂. The Southern Ocean returns to being a source of atmospheric $CO_2$ at approximately year 1400. At the end of the global warming experiment, the Atlantic and Southern Oceans are the major sink and source basins of atmospheric $CO_2$, respectively, as in the pre-industrial condition.

We also investigate the regional differences in uptake reduction due to global warming. In the first 500 years, approximately half of the total uptake reduction occurs in tropical and subtropical regions between 32.5° S and 32.5° N (Fig. 3c, and s4). The contributions of the high latitudes in the Northern and Southern Hemispheres have similar magnitudes. At a basin scale,

the contributions of the Atlantic and Pacific Ocean have similar magnitudes and that of the Indian Ocean is approximately half those contributions. These regional differences have also been found in similar long-term simulations by Plattner et al. (2001).

Figure 4 shows the zonal mean distribution of the salinity-normalized DIC (sDIC) in the global warming run. In the first 500 years, changes in sDIC from the pre-industrial condition decrease from the surface to the deep oceans. Significant sDIC changes in the deep water are observed in the North Atlantic. The shallower $CO_2$ invasion into the North Atlantic compared to the constant climate run (Fig. S4) is caused by the weaker and shallower AMOC. After year 1000, due to the enhanced AABW formation, AABW with a relatively high sDIC concentration intrudes into the deep North Atlantic below 2000 m. As
a result, the largest sDIC change in the deep water occurs in the North Atlantic at the end of the simulations. The smallest sDIC change is found in the deep North Pacific in accordance with the millennial ventilation timescale.

## 4 Effects of individual mechanisms on the reduction in oceanic CO2 uptake associated with global warming

The decrease in cumulative $CO_2$ uptake due to global warming projected by our model is consistent with previous centennial and millennial simulations using ESMs and EMICs. In this section, we disentangle the contributions of individual
mechanisms to $CO_2$ uptake reduction using three sets of sensitivity experiments (Abiotic, GW-base, and CTL-base; Tables 1 and 2). To compare our results with previous studies, we focus on changes in the $CO_2$ uptake in the first 500 years. The total reduction in the cumulative $CO_2$ uptake until year 500 is 402 PgC ("Total" in Table 2).

### 4.1 Abiotic experiments

In the abiotic experiments, the reduction in oceanic $CO_2$ uptake associated with global warming is caused by changes in the
ocean circulation and increases in the SST that reduce the global $CO_2$ uptake by 308 PgC and 140 PgC, respectively (Table 2). Weaker AMOC and AABW formations decrease the downward transport of $CO_2$ from the surface to the deep ocean, leading to a significant drop in the $CO_2$ uptake at high latitudes (Fig. 5a) and thus a reduction in sDIC in the deep ocean (Fig. 6a).

An increase in sDIC also occurs in the Antarctic intermediate water. As previously mentioned, the intensification of the Southern Hemisphere westerly winds enhances the mixing and overturning circulation of the Southern Ocean (Fig. 6a), increasing the $CO_2$ uptake and sDIC (Fig. 5a). An enhanced $CO_2$ uptake due to intensified westerly winds in the Southern Ocean has also been observed in previous studies using simple box models and coupled general-circulation models (Zickfeld et al., 2007; Matear and Lenton, 2008). A similar redistribution of carbon due to wind-stress-induced circulation changes are
found in Swart et al (2012), in which the Agulhas leakage and overturning circulation are linked.

The contribution of the SST increase to $CO_2$ uptake diminishes continuously from south to north. Biological changes calculated as the residual enhance $CO_2$ uptake at high latitudes and reduce it at low latitudes; the resulting global $CO_2$ uptake increase by 46 PgC due to biological changes. These global and regional contributions of individual mechanisms are consistent with previous estimations using abiotic experiments and other methods (Sarmient et al., 1998; Matsumoto et al., 2010).

### 4.2 GW-base and CTL-base experiments

### 4.2.1 Global change

The contributions of individual mechanisms to the uptake change estimated from the combination of the GW-base and CTL-base experiments are summarized in Table 2 and Figure 5b; averages of these experiments are used below. Changes in the organic matter cycle and in the solubility due to the SST increase are the dominant mechanisms for the global reduction in $CO_2$ uptake due to global warming (Fig. 5b). By year 500, these changes decrease the global carbon uptake by 276 PgC and 151 PgC, respectively (Table 2). A significant nonlinearity in the uptake change (from $-216$ PgC to $-336$ PgC) is found for the organic matter cycle. The ocean circulation change decreases the global $CO_2$ uptake by 6.5 PgC. The large differences relative to the abiotic simulation are discussed below (Section 4.3).

Enhanced $CaCO_3$ production reduces the surface alkalinity and therefore leads to an increase in the surface $pCO_2$. Changes in the $CaCO_3$ cycle reduce the $CO_2$ uptake by 14 PgC. The reduced sea ice cover allows air-sea gas exchange over a larger area and leads to a 15 PgC increase in the $CO_2$ uptake. Dilution of the surface salinity by 1 psu leads to a $pCO_2$ reduction of approximately 4% (Takahashi et al., 1993) and an increase in the $CO_2$ uptake. Enhanced evaporation at low latitudes increases $pCO_2$ resulting in an uptake reduction, whereas enhanced precipitation at high latitudes increases uptake. The global contribution of freshwater-flux changes is an uptake reduction of 2.5 PgC.

These significant contributions of changes in the organic matter cycle to the uptake reduction due to global warming and the indirect importance of circulation changes through nutrient transport and biological processes are consistent with Zickfield et al. (2008), who investigated the reduction in the oceanic $CO_2$ uptake due to freshwater hosing and AMOC weakening. However, these results are different from most previous studies (Sarmiento et al., 1998; Joos et al., 1999; Matear and Hirst, 1999; Plattner et al., 2001; Matsumoto et al., 2010), in which ocean circulation was the dominant mechanisms for reductions in oceanic $CO_2$ uptake. Below, we investigate what causes these greater biological effects and smaller circulation effects in our global warming simulations.

### 4.2.2 Effects of circulation change

The effects of circulation change display large regional variations (Fig. 5b). Similar to the abiotic experiments, reductions in $CO_2$ uptake and sDIC in the deep ocean (induced by weaker deep ocean circulation) occur at high latitudes, and sDIC is enhanced by increased westerly winds in the Southern Ocean. On the other hand, $CO_2$ uptake at low latitudes is enhanced by ocean circulation. We attribute this increase to weaker upper-ocean overturning and tropical upwelling (Fig. 7a). The tropical upwelling across 200m decreases from 38 Sv to 22.5 Sv in the first 200 years and continues to the end of the simulation (Fig. 7b). Weakening the equatorial upwelling reduces the upward transport of natural $CO_2$ from the ocean interior to the surface. The resulting enhanced $CO_2$ uptake at low latitudes increases sDIC in the subsurface water, especially in the Pacific Ocean (Fig. 6b). The importance of enhanced $CO_2$ uptake associated with weaker upper-ocean overturning is also suggested in decadal variability of oceanic $CO_2$ uptake (DeVries et al., 2017).

Enhanced $CO_2$ uptake induced by weaker equatorial upwelling and increased westerly winds cancel out the reduced $CO_2$ uptake associated with stratification and the weaker deep ocean circulation. Therefore, the effects of circulation changes are responsible for only a small fraction of the total reduction in the $CO_2$ uptake in our simulations. In the first 200 years, increases in $CO_2$ uptake caused by weaker equatorial upwelling and strengthened westerly winds are the dominant processes in the effect of circulation change (Fig. 8). Subsequently, the effects of weaker AMOC and AABW formations are enhanced and overcome those of the strengthened westerly winds and weaker equatorial upwelling. As a result, the $CO_2$ uptake is reduced by the circulation changes after year 450.

### 4.2.3 Effects of changes in the biological pump

Reduced nutrient supply into the euphotic zone due to circulation change and faster remineralization from seawater warming decrease the POC flux by 22% at a depth of 100m and 70% at a depth of 1000 m (Fig. 9a). The weaker carbon transport by the biological pump cannot maintain the vertical gradient of DIC between the surface and deep oceans. Excess DIC stored in the deep ocean by the biological pump is transported to the surface, therefore increasing surface $p$$CO_2$ and reducing $CO_2$ uptake. Reduced $CO_2$ uptake occurs at the low latitude and the high latitudes in the Northern Hemisphere (Fig. 5) according to changes in export production (Fig. 2g). Similarly, significant sDIC decreases occur in the North Atlantic, equatorial Indo-Pacific Ocean, and North Pacific (Fig. 9b). Reduction in $CO_2$ uptake induced by weaker biological pump are consistent with previous model simulations in which a significant atmospheric $CO_2$ increase is caused by a complete die-off of ocean life (Maier-Reimer et al., 1996; Sarmiento and Gruber, 2006).

Subsequently, the effect of the reorganization of the biological pump on the reduction in $CO_2$ uptake is decomposed into the effects of the temperature dependence of production and remineralization and the change in nutrient supply via the circulation change. In our model, the production and remineralization rate doubles for every 10°C increase in the seawater

temperature (Eppley, 1972). The warming-induced increase in production reduces the partial pressure of $CO_2$, therefore enhancing the oceanic $CO_2$ uptake. Enhanced remineralization of organic matter reduces the carbon transport into the deep ocean, leading to a lower $CO_2$ uptake. The lower nutrient supply due to stratification and slower thermohaline circulation decreases export production and therefore $CO_2$ uptake. The effect of the temperature dependence of the biological pump is calculated in GW-base and CTL-base ("Temperature for biology" in Table 2). The effect of the change in the nutrient supply is calculated as the residual ("Biology" − "Temperature for biology" as summarized in Table 2).

In the first 100 years, a warming-induced increase in production occurs more rapidly than an increase in remineralization because the surface waters are warming more rapidly than the subsurface. Therefore, the temperature dependence slightly increases the $CO_2$ uptake (Fig. 8). Subsequently, the effect of the enhanced remineralization overcomes that of the enhanced production; therefore the temperature dependence decreases the $CO_2$ uptake, consistent with the findings of Matsumoto (2007). By year 500, lower nutrient supply and temperature dependence decreases the $CO_2$ uptake by 241.5 PgC and 48.5 PgC, respectively (Table 2). Our results indicate that the reduction in the oceanic $CO_2$ uptake from biological change is mainly caused by reductions in nutrient supply via the circulation change, and that temperature effects on production and remineralization are not negligible.

The contribution of changes in the biological pump depends considerably on the base state (GW-base or CTL-base; 118 PgC difference while the total is 402 PgC) and mainly causes the non-linearity of feedback-mechanisms. This nonlinearity would be caused by difference in ocean circulation between GW-base and CTL-base. The enhanced $CO_2$ uptake associated with reduced upward transport of remineralized carbon partly cancel out the reduced $CO_2$ uptake due to lower export production. This process is larger in CTL-base than in GW-base because of slower deep ocean circulation in the latter. As a result, the contribution of changes in the biological pump is smaller in CTL-base than in GW-base. These nonlinearities need to be properly accounted for when separating the contribution of individual feedback-mechanisms to $CO_2$ uptake change.

## 4.3 Assessment of differences between the abiotic and CTL-base/GW-base experiments

The contributions of changes in the ocean circulation and the biological pump to the reduced $CO_2$ uptake differ between the abiotic and CTL-base/GW-base experiments even when using the same model and scenario, as suggested in previous studies (Sarmiento et al., 1998; Zickfeld et al., 2008; Matsumoto et al., 2010). Our results show that these different results are caused by the estimation methods used.

We attribute these different results to the vertical gradient of DIC under the pre-industrial condition of the abiotic and biotic experiments (i.e., CTL-base and GW-base) (Fig. 1). With regards to the circulation effect of the abiotic experiments, the enhanced $CO_2$ uptake associated with reduced upward transport of natural $CO_2$ from the deep ocean to the upper ocean is

underestimated because the vertical DIC gradient in the abiotic experiments is much smaller than in the biotic experiments. The effect of circulation change in the abiotic experiments primarily represents reduced $CO_2$ uptake due to reduced downward transport of anthropogenic $CO_2$ from the surface to deep ocean. In the CTL-base/GW-base experiments with a realistic vertical DIC gradient, the effect of the circulation change represents both a reduced upward transport of natural $CO_2$ and a reduced downward transport of anthropogenic $CO_2$. Therefore, the reduction in $CO_2$ uptake due to weaker deep-ocean circulation is larger in the abiotic experiments than in the CTL-base/GW-base experiments. An increase in $CO_2$ uptake associated with weaker equatorial upwelling is found only in the CTL-base/GW-base experiments.

In the abiotic experiments, the biological contribution is calculated as the residual. The enhanced $CO_2$ uptake owing to the reduced upward transport of natural $CO_2$ is included in the biological effect. This effect overcomes the reduced $CO_2$ uptake due to the weakening of the biological pump, leading to an enhanced $CO_2$ uptake due to the biological change. Therefore, the CTL-base/GW-base experiments are more useful for quantifying the contributions of circulation and biological change. We conclude that changes in the organic matter cycle and increases in the SST are the primary contributors to the reduction in $CO_2$ uptake due to global warming in our model. Although our experimental design differs from that used in some previous studies (e.g., Joos et al., 1999; Plattner et al., 2001), the identification of differences in feedback mechanism between all experimental designs is beyond the scope of this work and should be the subject of future research.

## 5 Summary

In this study, we investigated multi-millennium changes in the ocean carbon cycle under a quadrupling of the atmospheric carbon dioxide using an AOGCM and an offline biogeochemical model. At the end of the simulation, the cumulative $CO_2$ uptake is 2028 PgC and global warming reduces the uptake by 30%; this primarily occurs in the tropical and subtropical regions. These projected global and regional changes in oceanic $CO_2$ uptake are similar to those in previous long-term simulations using EMIC (Plattenr et al., 2001; Schmittner et al., 2008).

To isolate and quantify the individual mechanisms responsible for the reduction in the oceanic $CO_2$ uptake due to global warming, we conducted three sets of sensitivity experiments based on previous studies. Our results are consistent with those previous studies: in abiotic experiments, the primary contributors are changes in the ocean circulation and reduced solubility due to sea-surface warming (Sarmient et al., 1998; Matsumoto et al., 2010), while in the CTL-base/GW-base experiments, the primary contributors are changes in the organic matter cycle via ocean circulation change and sea-surface warming (Zickfeld et al., 2008). It is conceivable that the differences in results between these sensitivity studies are caused by the vertical gradient of DIC under pre-industrial conditions, i.e., smaller DIC gradients in abiotic experiments and realistic DIC gradients in CTL-base/GW-base experiments (Fig. 1). In the abiotic experiments, the circulation effects are underestimated with respect to enhanced $CO_2$ uptake associated with reduced upward transport of natural $CO_2$ from the deep ocean to the

upper ocean. On the other hand, in the CTL-base/GW-base experiments, the effects of the circulation changes include both reduced upward transport of natural $CO_2$ and a reduced downward transport of anthropogenic $CO_2$. Therefore, the CTL-base/GW-base experiments are more useful for quantifying the individual feedback mechanisms.

The reduced $CO_2$ uptake associated with weaker deep ocean circulation and stratification is counteracted by the enhanced uptake due to weaker equatorial upwelling and strengthened subpolar westerly winds in the Southern Hemisphere. The weakening of the biological carbon export resulting from circulation change and the enhanced remineralization owing to seawater warming diminishes the DIC gradient between the surface and deep ocean, leading to a significant reduction in oceanic $CO_2$ uptake. Our results suggest that changes in the biological pump and solubility are the primary drivers of

reductions in oceanic $CO_2$ uptake associated with global warming.

Circulation plays a small direct role on changes in oceanic carbon uptake due to global warming, but a large indirect role through nutrient transport and the biological pump. On a centennial timescale, the weakening of AMOC and AABW formation in our simulation is consistent with the results of CMIP5 models (Weaver et al., 2012; Heuzé et al., 2015).

Changes in oceanic carbon uptake and reductions in export production due to global warming are in close agreement with current simulations using ESMs, as mentioned in Section 3. Therefore, we speculate that similar contributions to the reduction in $CO_2$ uptake from changes in ocean circulation and biological processes will be found in other models. On the other hand, the longer-term responses of AMOC and AABW are very uncertain. The recovery and overshoot of the AABW formation that occurred after 1000 years in our simulation have not been reported in the previous multi-millennium

simulations (Schmittner et al., 2008, Li et al., 2013). To fully assess the robustness of millennial-scale reductions in oceanic $CO_2$ uptake due to global warming, simulations using other recent GCMs and ocean biogeochemical models are required.

Although our results show that changes in the biological pump are crucial for reductions in oceanic $CO_2$ uptake due to global warming, several biological processes were not considered in our model. A reduction in the saturation levels of calcium

carbonate in seawater associated with $CO_2$ invasion into the ocean can have adverse effects on marine calcifying organisms. The subsequent reduction in $CaCO_3$ production would increase the surface ocean alkalinity and therefore enhance oceanic carbon uptake (Riebesell et al., 2000), but its modelled impact has been very different in previous studies (Heinze, 2004; Gehlen et al., 2007; Ridgwell et al., 2007; Matsumoto et al., 2010). In addition, an increase in biotic carbon-to-nitrogen drawdown in response to $p$$CO_2$ changes can enhance oceanic $CO_2$ uptake (Oschlies et al., 2008). Reduced seawater viscosity

in a warmer ocean accelerates particle sinking speed, therefore increasing the efficiency of the biological pump and enhancing oceanic $CO_2$ uptake (Bach et al., 2012; Taucher et al., 2014). These effects remain poorly understood but could potentially have a large impact on oceanic carbon uptake and atmospheric $CO_2$ concentration on a multi-centennial to millennial timescale. The inclusion of these processes in models may be required for a more comprehensive understanding of the response of the ocean carbon cycle to global warming.

**Acknowledgements**

Data are freely available upon request from the corresponding author (akitomo@jamstec.go.jp). The authors thank two anonymous reviewer and Neil Swart for constructive reviews. This research was supported by the Environment Research and Technology Development Fund (S-10) of the Ministry of the Environment, Japan, and was also supported by the

"Integrated Research Program for Advancing Climate Models (TOUGOU program)" from the Ministry of Education, Culture, Sports, Science and Technology (MEXT), Japan. AOGCM simulations were carried out on the JAMSTEC Earth Simulator. The simulations with offline biogeochemical model were performed using the Fujitsu PRIMEHPC FX10 System in the Information Technology Center, University of Tokyo.

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

**Figures**

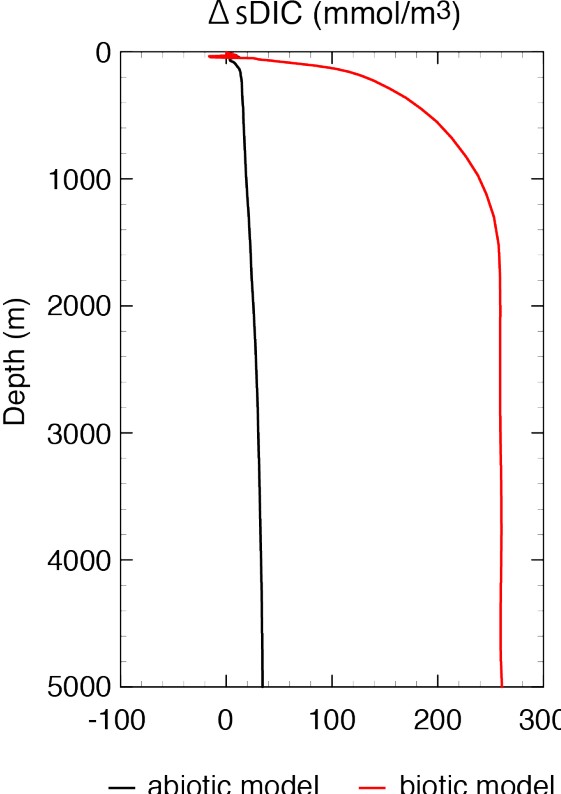

**Figure 1.** Vertical profile of the anomaly of salinity-normalized DIC from the surface under the pre-industrial condition. The black and red lines show the abiotic and biotic models, respectively.

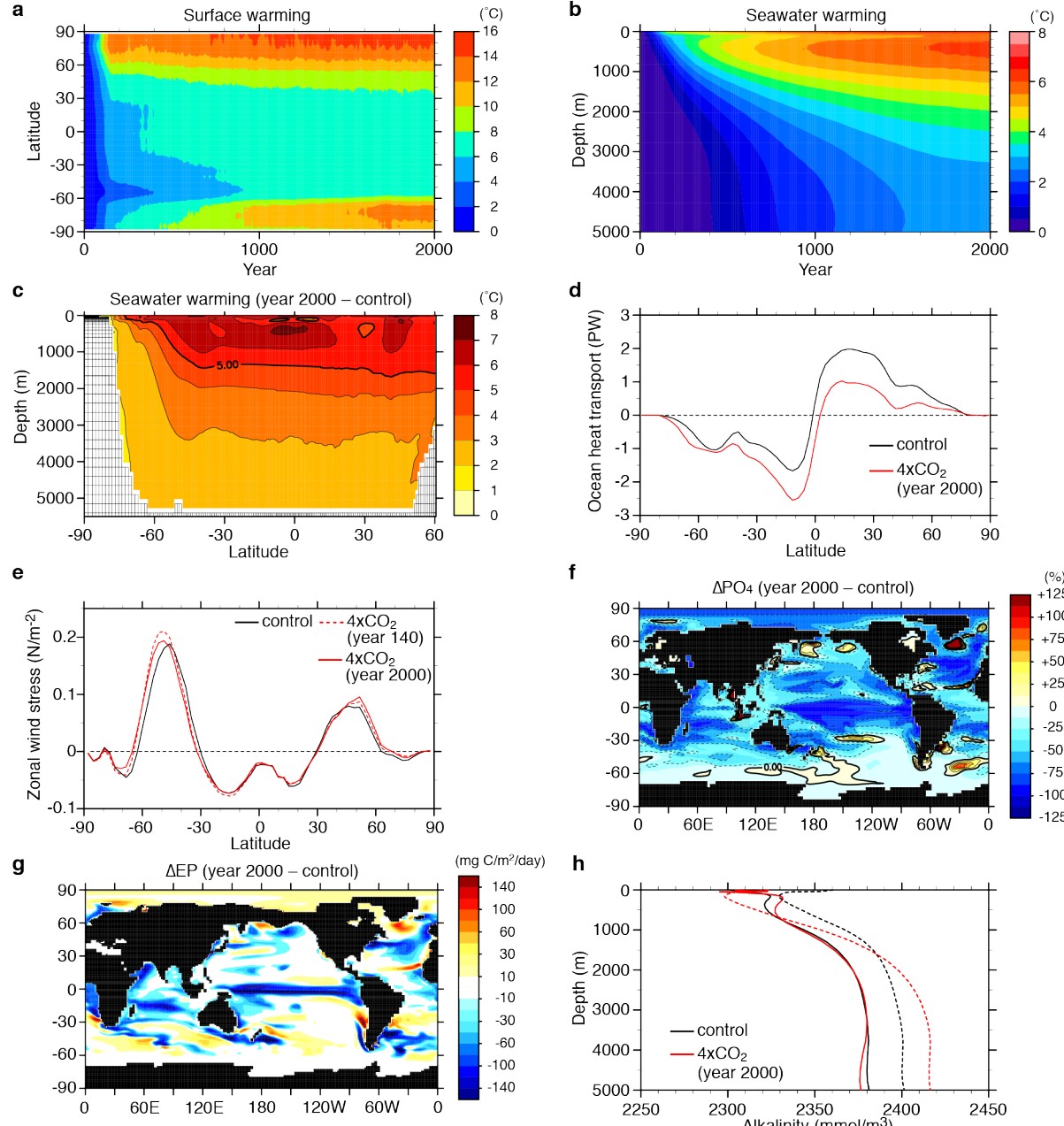

**Figure 2.** Changes in the physical and biological variables in the global warming experiment. Hovmöller diagrams of (a) the zonally averaged anomaly of the surface temperature and (b) the horizontally averaged anomaly of the seawater temperature. (c) Zonal mean changes in the seawater temperature between the control run and the end of the global warming experiment (year 2000). (d) Northward ocean heat transport for the control (black) and the end of the global warming runs (red). (e) Zonal mean wind stress for the control run (black) and at the time of $CO_2$ quadrupling (years 140 and 2000: red dashed and solid, respectively). Changes in (f) the surface $PO_4$ (averaged over the top 50 m) and (g) the export production at the end of the global warming run. (h) Vertical profile of the alkalinity (solid lines) and salinity-normalized alkalinity (dashed lines) for the control (black) and global warming (red) runs.

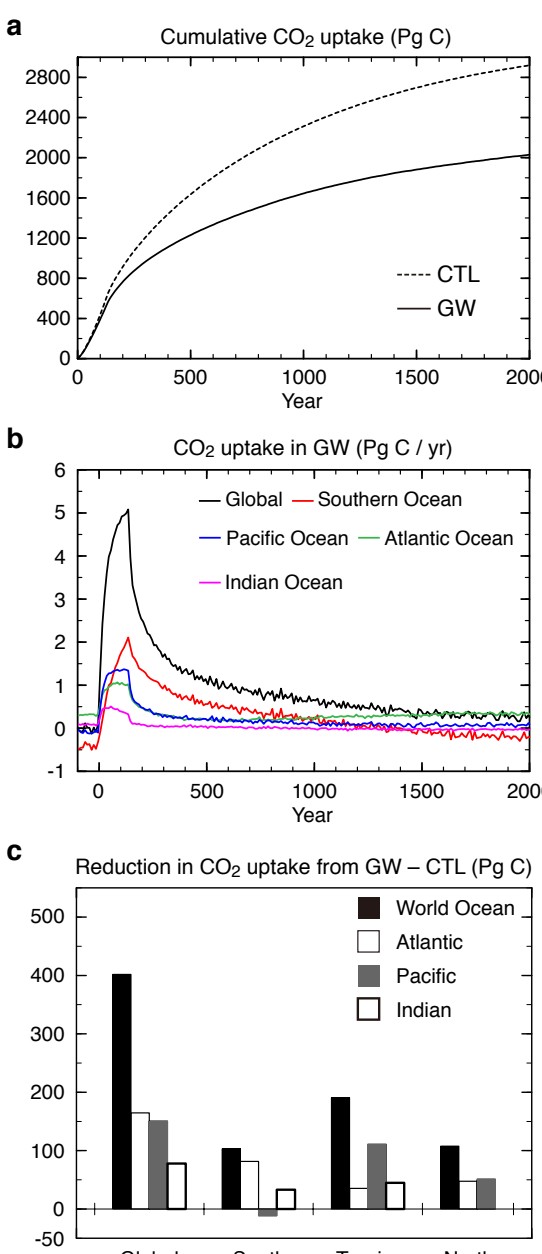

**Figure 3.** (a) Time series of the cumulative oceanic $CO_2$ uptake for CTL (dashed line) and GW (solid line). (b) Time series of the annual mean values of the oceanic $CO_2$ uptake for the global ocean (black) and different ocean basins (colors) in GW. (c) Reduction in the oceanic $CO_2$ uptake due to global warming for the global ocean and three ocean basins at 500 years (GW–CTL). The contributions of the global ocean and individual ocean basins are separated into three regions (South: south of 32.5° S; Tropics: 32.5° S to 32.5° N; North: north of 32.5° N). The positive and negative values represent uptake reductions and uptake increases, respectively.

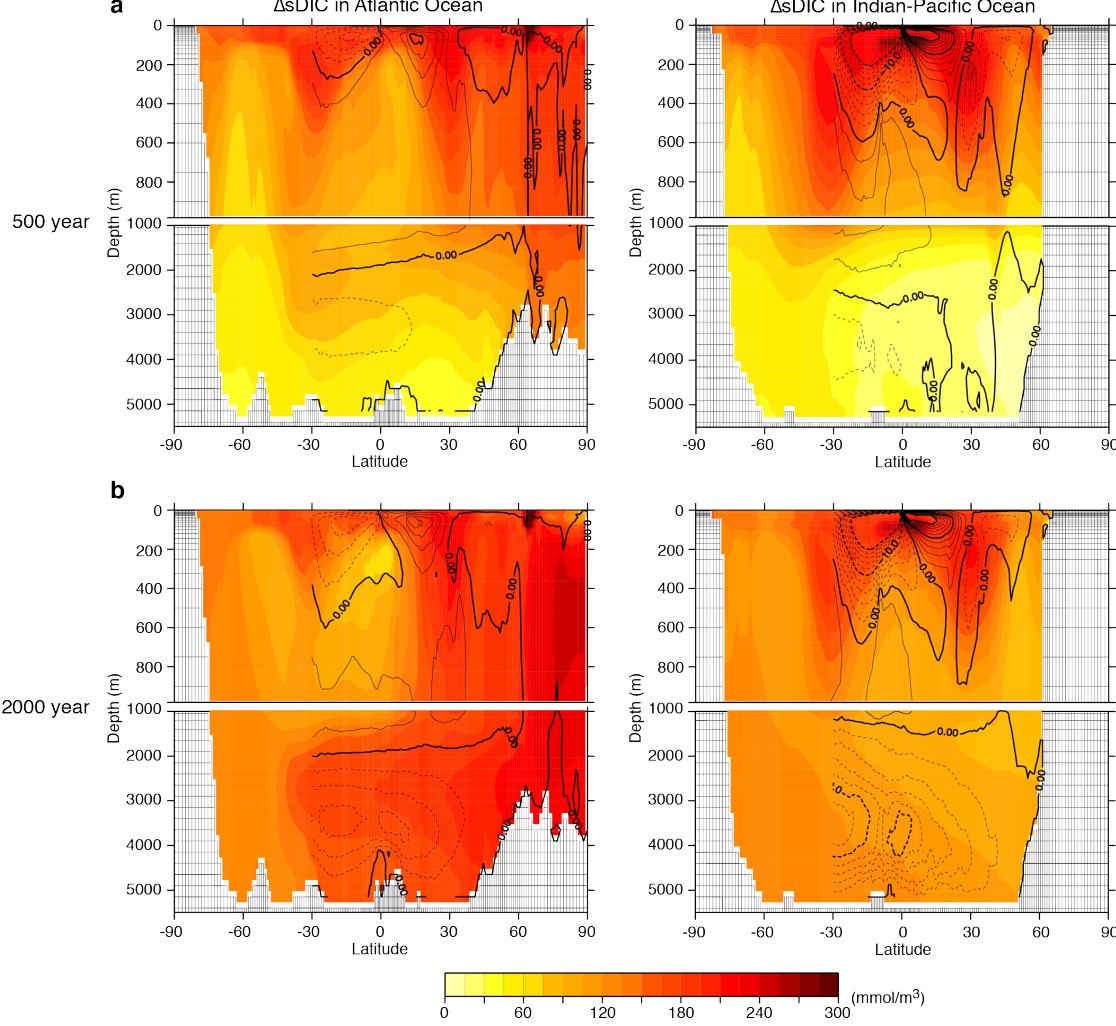

5  **Figure 4.** Zonal mean distribution of changes in the salinity-normalized DIC (colors) and meridional overturning stream function (contours) for (a) 500 years and (b) 2000 years in the global warming run (GW). The left and right panels show the Atlantic and Indo-Pacific Oceans, respectively; contour interval is 2 Sv.

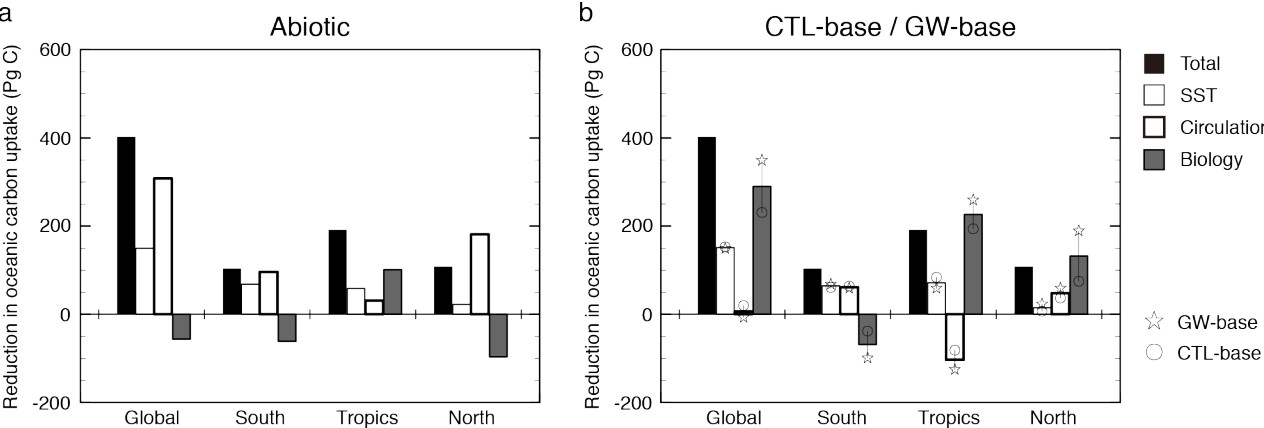

**Figure 5.** Contributions of the primary mechanisms to the reduction in the oceanic $CO_2$ uptake due to global warming at 500 years for (a) the abiotic experiments and (b) the CTL-base/GW-base experiments. Total change and effects of SST, circulation, and biology are calculated as summarized in Table 2. The biological effect in the abiotic experiments is calculated as the residual. The biological effect in the CTL-base and GW-base represents the effect of organic matter cycle. Contributions of the individual mechanisms are separated into three regions (South: south of 32.5° S; Tropics: 32.5° S to 32.5° N; North: north of 32.5° N). In panel (b), bars refer to the averages of CTL-base and GW-base.

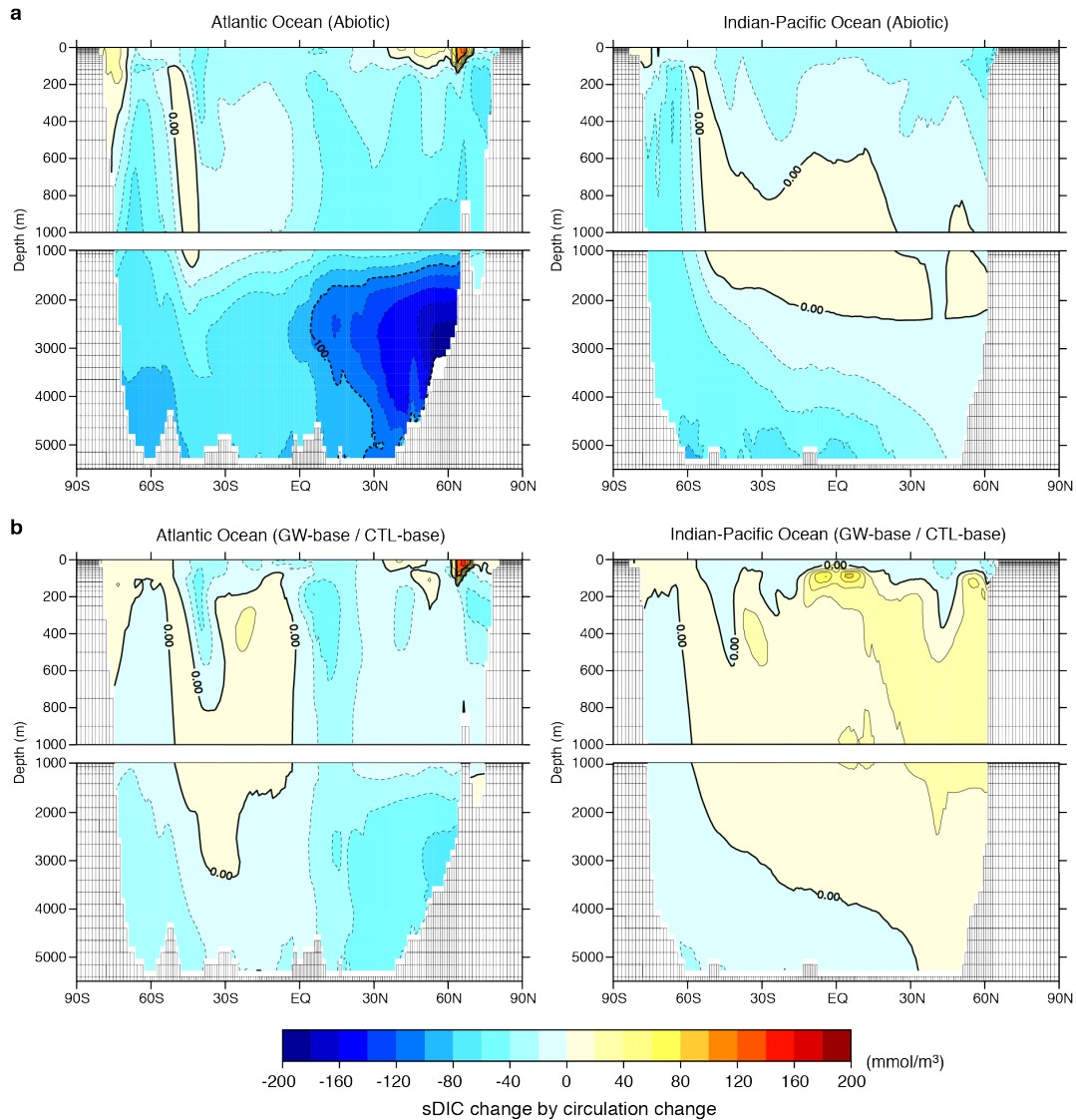

**Figure 6.** Zonal mean changes in the salinity-normalized DIC induced by circulation changes for (a) the abiotic experiments and (b) the CTL-base/GW-base experiments at 500 years. The left and right panels show the Atlantic and Indo-Pacific Oceans, respectively.

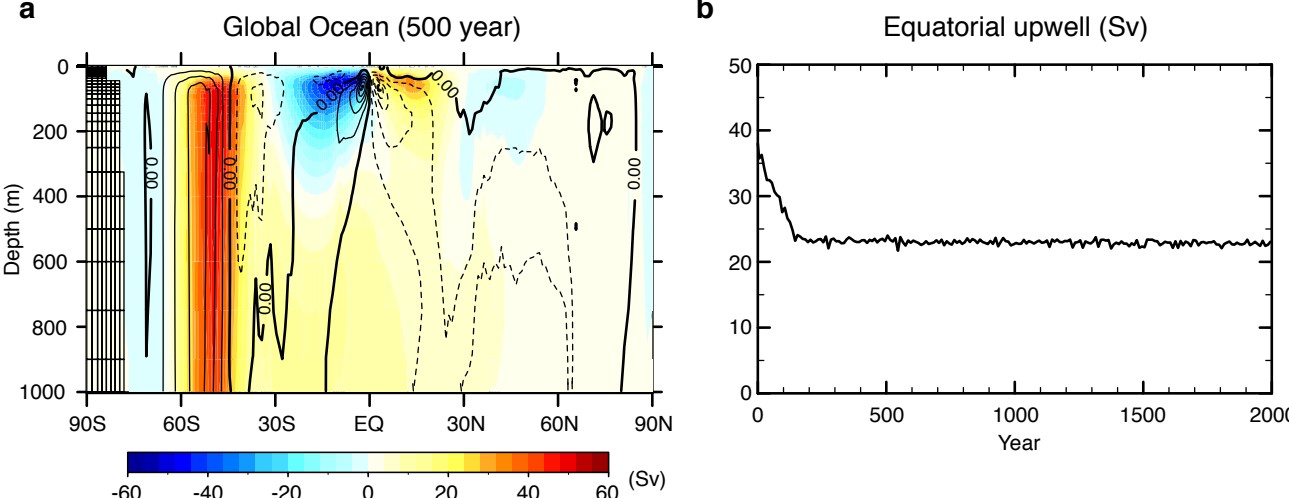

**Figure 7.** Global upper-ocean overturning circulation. (a) Meridional overturning stream function at year 500 (colors). The positive numbers represent clockwise circulation, and the negative numbers represent counterclockwise circulation. The contours show the differences between meridional overturning stream function in year 500 and the pre-industrial condition.
5   The contour interval is 4 Sv. (b) Time series of global equatorial upwelling (through 200 m, between 10˚ S and 10˚ N)

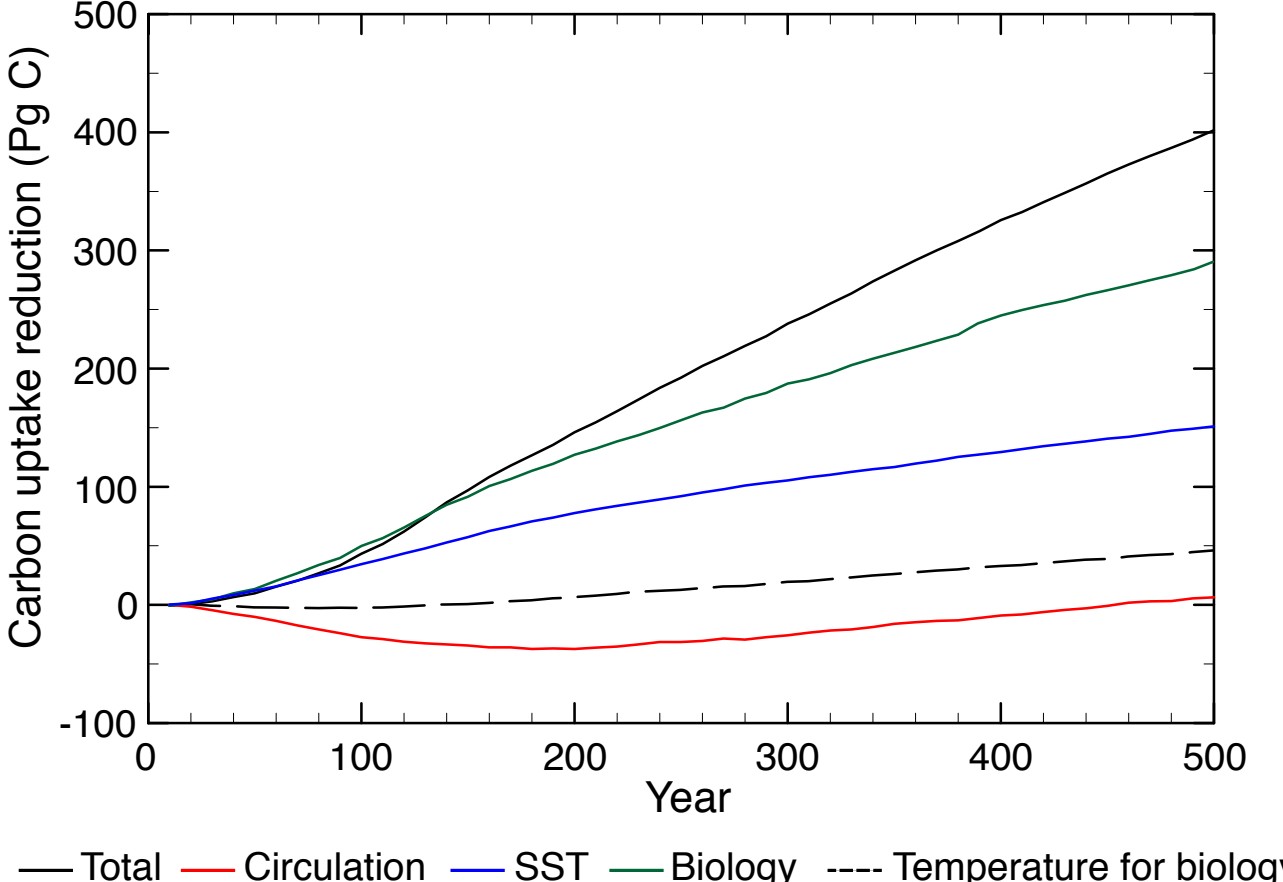

**Figure 8.** Time series of the contributions of primary mechanisms to the reduction in the oceanic $CO_2$ uptake in the CTL-base/GW-base experiments. The positive and negative values represent uptake increases and decreases, respectively. The contributions of circulation (red), biology (green), and SST (blue) and the temperature dependences of production and remineralization (black dashed line) are illustrated. The quantification methods for each mechanism are summarized in Table 2.

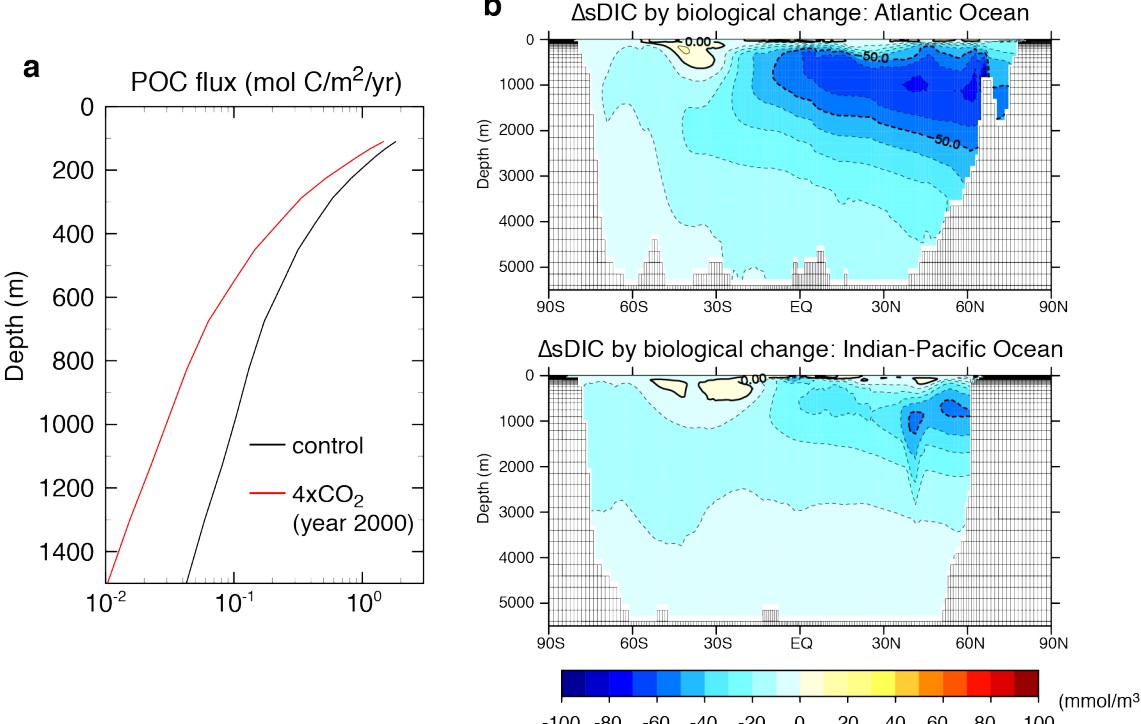

**Figure 9.** Contribution of the reorganization of the organic matter cycle. (a) Vertical distribution of the POC flux for the control (black) and global warming runs (red). (b) Zonal mean changes in the salinity-normalized DIC induced by changes in the organic matter cycle in the CTL-base/GW-base experiments at 500 years. The top and bottom panels show the Atlantic and Indo-Pacific Oceans, respectively.

| | Experiments | Changing mechanisms | Cumulative 500year | uptake (Pg C) 2000year | SST | Dilution | Circulation | Organic matter cycle | CaCO$_3$ cycle | Sea ice |
|---|---|---|---|---|---|---|---|---|---|---|
| 1 | CTL_bio | – | 1629 | 2888 | CTL | CTL | CTL | CTL | CTL | CTL |
| 2 | GW_bio | all | 1227 | 2028 | GW | GW | GW | GW | GW | GW |
| 3 | CTL_abio | – | 1819 | | CTL | CTL | CTL | – | – | CTL |
| 4 | GW_abio | SST circulation | 1371 | | GW | GW | GW | – | – | GW |
| 5 | GW_abio_SST | SST | 1511 | | CTL | GW | GW | – | – | GW |
| 6 | GW_bio_SST | SST | 1376 | | CTL | GW | GW | GW | GW | GW |
| 7 | GW_bio_fw | freshwater flux | 1230 | | GW | CTL | GW | GW | GW | GW |
| 8 | GW_bio_circ | circulation | 1220 | | GW | GW | CLT | GW | GW | GW |
| 9 | GW_bio_om | organic matter cycle | 1563 | | GW | GW | GW | CTL | GW | GW |
| 10 | GW_bio_ca | CaCO$_3$ cycle | 1240 | | GW | GW | GW | GW | CTL | GW |
| 11 | GW_bio_si | sea ice | 1211 | | GW | GW | GW | GW | GW | CTL |
| 12 | GW_biotemp | temperature for biology | 1278 | | GW | GW | GW | GW* | GW* | GW |
| 13 | CTL_bio_SST | SST | 1476 | | GW | CTL | CTL | CTL | CTL | CTL |
| 14 | CTL_bio_fw | freshwater flux | 1627 | | CTL | GW | CTL | CTL | CTL | CTL |
| 15 | CTL_bio_circ | circulation | 1609 | | CTL | CTL | GW | CTL | CTL | CTL |
| 16 | CTL_bio_om | organic matter cycle | 1413 | | CTL | CTL | CTL | GW | CTL | CTL |
| 17 | CTL_bio_ca | CaCO$_3$ cycle | 1614 | | CTL | CTL | CTL | CTL | GW | CTL |
| 18 | CTL_bio_si | sea ice | 1643 | | CTL | CTL | CTL | CTL | CTL | GW |
| 19 | CTL_biotemp | temperature for biology | 1583 | | CTL | CTL | CTL | CTL** | CTL** | CTL |

*Pre-industrial seawater temperature is used for calculating biological term in GW_biotemp.

**Seawater temperature of GW_bio is applied for calculating biological term in CTL_biotemp.

**Table 1:** Description of the model experiments and results for oceanic CO$_2$ uptake.

| Mechanisms | Abiotic | uptake change (Pg C) | GW-base | uptake change (Pg C) | CTL- base | uptake change (Pg C) | average of GW-base and CTL-base (Pg C) |
|---|---|---|---|---|---|---|---|
| Total | $(2)-(1)$ | -402 | $(2)-(1)$ | -402 | $(2)-(1)$ | -402 | -402 |
| SST | $(4)-(5)$ | -140 | $(2)-(6)$ | -149 | $(13)-(1)$ | -153 | -151 |
| Freshwater | − | − | $(2)-(7)$ | -3 | $(14)-(1)$ | -2 | -2.5 |
| Circulation | $(5)-(3)$ | -308 | $(2)-(8)$ | 7 | $(15)-(1)$ | -20 | -6.5 |
| Sea ice | − | − | $(2)-(11)$ | 16 | $(18)-(1)$ | 14 | 15 |
| Biology | (Total) − (SST) − (Biology) | 46 | (Organic matter cycle) + (CaCO3 cycle) | -349 | (Organic matter cycle) + (CaCO3 cycle) | -231 | -290 |
| Organic matter cycle | − | − | $(2)-(9)$ | -336 | $(16)-(1)$ | -216 | -276 |
| CaCO3 cycle | − | − | $(2)-(10)$ | -13 | $(17)-(1)$ | -15 | -14 |
| Temperature for biology | − | − | $(2)-(12)$ | -51 | $(19)-(1)$ | -46 | -48.5 |
| Nutrient supply | − | − | (Biology) − (Temperature for biology) | -298 | (Biology) − (Temperature for biology) | -185 | -241.5 |
| residual | | | | 76 | | -10 | 33 |

**Table 2:** The contributions of individual mechanisms to the reduction in the $CO_2$ uptake due to global warming in the first 500 years.