# Peer review of "Long-term response of oceanic carbon uptake to global warming via physical and biological pumps"

_Biogeosciences, 2017_

## Referee Comment (RC1) · N.C. Swart (Referee) · 18 Dec 2017

**General comments**

This paper seeks to understand the source of carbon climate feedbacks arising in the ocean on multi-centennial timesscales. This is an important question in the Earth System Modelling community, including for understanding future climate change, and interpreting carbon budgets. The authors use a well thought out experimental design to quantify the sensitivity of different aspects of the ocean carbon cycle (e.g. biology, circulation, solubility etc) to climate change. The approach is based on previous work,

but fairly novel in this particular application. The paper is well organized and written, and the results, including the graphics are clear. Most uncertainties are addressed and the results are placed in the context of previous work. I thoroughly enjoyed this paper. Almost every time I had a question it was answered in the follow sentence or section. Overall I assess the quality as very high, and I recommend publication. I don't have any major issues. I do have some comments which I think could help to clarify the paper and address the few lingering questions that I did have.

**Specific comments**

- The authors describe a decreasing ocean $CO_2$ uptake under global warming, and attribute this in large part to a reduction in export production. However previous literature (e.g. de Vries et al. [2012], Marinov et al. [2008] and references therein) has shown that ocean $CO_2$ uptake is not directly tied to export production (as one might guess), but rather to the so called "efficiency of the biological pump". Please clarify how export production, biological pump efficiency and carbon uptake relate in this study. Specifically, is it really export production which is important - and if so why is this different from the above literature?

- The authors force the offline ocean biogeochemical model with monthly mean fields from the AOGCM (including for insolation, velocity, temperature, salinity etc). This means that much variability is being averaged over, including the diurnal cycle, synoptic scale variability and so on. There is a known sensitivity of ocean model response to forcing frequency. Obviously forcing at a higher frequency means more data, and is more expensive. But please discuss how the results might be sensitive to the forcing frequency. I don't necessarily need any more experiments, just a clear caveat on this point.

- Circulation plays a small direct role, but a large indirect role through nutrient transport. The circulation changes are large (and mostly consistent with expectations). In various parts of the manuscript, the authors do a good job of comparing their results to those from CMIP and other studies. If possible it would be interesting to know how the MIROC simulated circulation changes under 4xCO2 compare to other CMIP models. More generally a comment on how sensitive the results are to uncertainties, for example in the climate model response to increasing CO2, would be helpful. (I note the authors do discuss the need for similar studies using different models, but the reasons for this could be fleshed out).

**Technical comments and typos (by pg and ln)**

pg 1 / Abstract:

ln 8: "accelerate an increase in CO2" - Is there really an "acceleration". I'm not sure that this is the right word. I think just "decrease oceanic carbon uptake and therefore increase atmospheric CO2 and global warming" would sound better and be more accurate.

ln 14: "...first 140 years (at year 2000)" - the meaning of this because clear later when reading the methods, but this could be a little confusing in the abstract, because readers do not know at that point what experiment you are conducting. For example, on first reading I was thinking "calendar year 2000".

ln 19: "...gradient of DIC substantially" - add a comma after "DIC"

ln 23-4: "uptake through natural carbon cycle" - suggest removing "natural carbon cycle". I don't think this is needed.

pg 2:

ln 5-6: "...long-term evolution of climate systems with slow response times..." -> "...long term evolution of climate system components with a slow response time..." (i.e. there is only one climate system, which is made up of many components).

ln 10: "accelerating the rate of CO2 accumulation" - again I'm not sure if "accelerating" is accurate? Maybe just "increasing CO2 accumulation in the atmosphere".

ln 13: "primarily alter"...delete "primarily". There are only the natural and anthropogenic CO2 cycles.

ln 15-16: Another good study to reference is Randerson et al. (2015). They show that ocean carbon feedbacks become larger than land carbon feedbacks, but only on very long time scales. There is a nice tie in with this work.

ln 15-20: I suggest mentioning here that you will explain later why those studies came to that conclusion (and are different from yours).

ln 25 "However the contributions"...suggest deleting "However". This sentence is not really a continuation of the previous sentence.

ln 26-27: There are no studies doing this breakdown for CMIP5?

ln 28 "with AOGCM" -> "with an AOGCM"

pg 3:

ln 3 "using AOGCM" -> "an AOGCM".

ln 13 "with MIROC 4m AOGCM" -> "with the MIROC 4m AOGCM"

ln 27-28 "according to AOGCM climate simulations" - I got what you meant, but this could be clearer. Maybe something like "following the physical evolution of AOGCM climate simulations", or "forced by output from AOGCM climate simulations".

pg 4:

ln 11: "setting flux" -> "settling flux"

ln 16-18: "we confirmed..." - I found this confusing. At the bottom of page 3, it says that salinity is specified from the AOGCM simulations - but here you are saying you are using salinity from the offline simulation to validate against the AOGCM simulation. Something is missing. Do you simulate a passive salinity tracer in the offline model, to compare against the "online" salinity in the AOGCM? Please clarify.

ln 25-31: Just noting that the comparison is between a pre-industrial simulation and modern observations. This could have some impact. Are you using GLODAP estimated PI DIC and ALK to compare against? Not a big deal but worth clarifying.

pg 5:

ln 5-7 "This model does not include..." - it seems like these sentences belonged in section 2.2 to me. They are about the model, not the experiment.

ln 9 "We conducted additional experiments"...these were only run for 500 years, right? Maybe worth mentioning here.

ln 9-20: It is mentioned briefly below, but I think it is worth mentioning clearly here at the outset that the experimental design assumes linearity of the feedbacks.

ln 23: "and oceanic interior temperature and salinity". When I thought about the experimental design - as far as I can tell these interior T and S values are not used for anything in the offline model for this particular experiment, since the organic matter cycle is specified. The SST is, I believe, still be specified as GW. If this is all true, I would just remove the mention of "interior T and S values", since it is not relevant, and could be confusing. If these values are used for something, please clarify.

pg 6:

ln 12 :"after the summary of the global mean" - a bit confusing as written. Maybe "...and ocean biogeochemical variables. A full summary of the global mean changes is reported in..."

pg 7:

ln 2 / fig 1 e: I suggest you add the line for wind stress at year 2000 to Fig 1e (most other panels in fig. 1 are showing a year 2000 result). It would be helpful to see the recovery.

ln 6-15: $PO_4$ is shown, but what about $NO_3$? More generally, the paper discusses export production in general, but does not mention how diazotrophs and "other" phytoplankton react?

pg 8:

ln 6: "...during constant atmospheric $CO_2$..." - I would include the year 140, as in "...constant atmospheric $CO_2$ after year 140..." for clarity.

ln 27-33: I was interested in this section, and would like to see more spatial information. If possible, it would be really nice to see a Hovmoller, like Fig 1a, but for $CO_2$ uptake/flux anomaly (of GW - CTL) (maybe in the SI).

pg 9:

ln 23-24: I suggest you reference these "uptake change" numbers back to table 2.

pg 10:

ln 19-23: Le Quere et al 2008 claim that the westerly wind increase is reducing Southern Ocean $CO_2$ uptake (i.e. the opposite of what is being said here). Therefore, it is strange to cite as evidence without further explanation. I suggest it would be better to reference the Zickfeld et al. response to Quere et al. (who show that the $CO_2$ uptake response to wind changes is time-scale dependent).

The effect of circulation change on sDIC (Fig 5) is essentially a redistribution of carbon from the Atlantic to the Pacific. Interestingly, we saw a similar redistribution due to to wind stress induced circulation changes in Swart et al. (2012), which we linked back to changes in the Agulhas leakage and overturning circulation.

Figures:

1. e : please add line for year 2000

3. The colorbar is not perceptually uniform, which makes it hard to determine where large changes have actually occurred. Please consider using a perceptually uniform colorbar.

6. Caption "Global upper-ocean" - fix typo

**References**

DeVries, T., F. Primeau, and C. Deutsch (2012), The sequestration efficiency of the biological pump, Geophys. Res. Lett., 39, L13601, doi:10.1029/2012GL051963.

Randerson, J. T., K. Lindsay, E. Munoz, W. Fu, J. K. Moore, F. M. Hoffman, N. M. Mahowald, and S. C. Doney (2015), Multicentury changes in ocean and land contributions to the climate-carbon feedback, Global Biogeochem. Cycles, 29, 744–759. doi:10.1002/2014GB005079.

Marinov, I., A. Gnanadesikan, J. L. Sarmiento, J. R. Toggweiler, M. Follows, and B. K. Mignone (2008), Impact of oceanic circulation on biological carbon storage in the ocean and atmospheric pCO2, Global Biogeochem. Cycles, 22, GB3007, doi:10.1029/2007GB002958.

Swart, N.C. and J.C. Fyfe (2012) Ocean carbon uptake and storage influenced by wind bias in global climate models, Nature Climate Change 2, 47–52, doi:10.1038/nclimate1289

ZICKFELD, K. JOHN C. FYFE, MICHAEL EBY, ANDREW J. WEAVER Comment on "Saturation of the Southern Ocean CO2 Sink Due to Recent Climate Change",SCIENCE01 FEB 2008 : 570

---

## Referee Comment (RC2) · Anonymous Referee #2 · 22 Dec 2017

The manuscript by Yamamoto et al explores through a large suite of experiments under fixed atmospheric concentrations the role physical changes in climate play on ocean carbon uptake. Their conclusions suggest, in contrast to other papers, that the change of circulation dominate the response. It took me a little while to get into this paper, but once there I enjoyed the paper much and really appreciate the larger number of simulations that went into this work - thank you. Overall this is well conceived and executed piece of work, that will be of interested to a wide readership. I do have some minor comments that I feel once addressed would strengthen the paper, otherwise I am happy to recommend this paper for publication.

Minor Comments:

1. The authors predicate the study on global warming, and state that global warming will decrease ocean carbon uptake. However in the present day, as $CO_2$ levels continue to rise - the ocean will take up carbon at a rate proportional to this i.e. gradient driven. I do understand in this study, if we assume fixed $CO_2$ levels then this supposition is correct, but I do think this needs to clarified in the text.

2. The study puts more heat and carbon into the ocean over a much shorter period than under CMIP3/5 change changes runs, even the business-as-usual scenario; this of course has implications for where the heat and carbon are stored. As the authors make a number comparison to these climate change runs - could they comment on what the implications of this maybe - perhaps on the timing of events e.g. sinks to sources etc, and whether its a fair comparison?

3. The experimental methods section is super critical to this paper, however I needed to read this at least 5 times to be really clear. I recommend that the authors break up the 3rd paragraph to make it more accessible

4. The study uses offline simulations, which make sense, could the authors comments on whether on or offline makes much difference - given the challenges of capturing short-term processes in the fields needed to run the model. I am sure that they have tested this somewhere, and if not it should be acknowledged.

5. The timescales calculated in the paper are based on a fixed atmospheric concentrations. In the real world i.e. driven by emissions, the ocean carbon uptake would significantly slow as the gradient between the ocean and atmosphere decreases. I think this probably needs to be mentioned in the discussion, as do the implications for timing of changes.

6. Otherwise some minor typos etc need to be addressed, but I am sure they will be picked in the proofs.

---

## Referee Comment (RC3) · Anonymous Referee #3 · 23 Dec 2017

The authors study long term ocean carbon cycle feedbacks over a time horizon of 2000 years by using an offline ocean biogeochemistry model driven by climate model output. By using different combinations of output fields from a control simulation (no global warming) and a global warming simulation, they separate the carbon cycle feedback into components originating from SST-changes, circulation changes, changes of the biological pump, and a few others. They find that changes in the biological pump contribute most to the carbon uptake reduction under climate change followed by solubility changes. The authors claim that this finding is "contrary to most previous studies".

The manuscript is clearly within the scope of Biogeosciences. The main conclusions,

however, are partly inconsistent and not well enough supported by the results. Also, the manuscript as it stands now, it is not very novel. Many similar studies on ocean carbon-climate feedbacks have been published during the past 20 years, most of them with simpler models. However, the authors do not convincingly make the point as to why significantly different results could be expected because of enhanced model complexity. There are (or potentially are) interesting new aspects in the present study, but the authors do not elaborate these (see below).

Major points:

1) The statement that the results are "contrary to most previous studies" is not convincingly supported by the results presented in this manuscript. Since the experimental set-up is different from (most of the) previous studies, it remains unclear what the effect of these differences might be. This is briefly discussed in section 5, following speculations (page 12, lines 20-30) about why models in previous studies possibly gave different results. These speculations are not convincingly supported by the results or the cited literature either. In my opinion it is most likely that differences in the experimental set-up explain much of the differences. The authors follow Zickfeld et al. (2008) in designing their experiments, and use the tendencies of DIC and ALK due to biological production/remineralisation from the CTL-experiment in the GW-experiment (and vice versa) to determine the effect of biology on CO2 uptake. This mimics pre-industrial organic matter and CaCO3 production/remineralisiation under a reduced circulation. I find this design questionable, since it weakens the upward transport of remineralised carbon and nutrients (leading to enhanced C-uptake), but at the same time keeps the export production at pre-industrial levels (leading also to enhanced uptake). The experiment design used in some of the previous studies (Joss et al. 1999, Plattner et al. 2001) is different: Here, archived pre-industrial surface sPO4 and sALK fields are used in a global warming simulation to separate the "effect of biology". If I am not mistaken, the effect of reduced upwelling of DIC is cancelled out in this experiment. Other studies (Sarmiento et al. 1998, Matsumoto et al. 2010) use abiotic experiments. In order

to demonstrate that the feedback-mechanisms are really substantially different from previous studies, the authors would need to quantify the differences arising due to the different experimental set-up (or different interpretations of the "biological effect"). This could be done by running additional sets of experiments following the design of previous studies. A discussion of which experimental set-up or definition of "biological pump contribution" is more useful or correct should also be provided. The authors state in the abstract that "quantifications of the contributions from different processes to the overall reduction in ocean uptake are still unclear". Instead of adding to the confusion they could take the opportunity to assess what the experimental set-up in different studies contributes to this. This would also be a novel and useful contribution to the field.

In this context it would be also useful to discuss the limitations of the approaches to separate the feedback-mechanisms in a non-linear system. The authors have already performed two sets of experiments (GW-base and CTL-base), which could serve this purpose. Results from these sets of experiments are presented in Table 2, but are only mentioned in one brief sentence (page 9, lines 24-25) in the manuscript. Particularly, for the "Biology"-contribution, the authors find a considerable dependence on the base state (GW or CTL; 118 PgC difference while the total is 402 PgC). An explanation for this would be useful. Do the authors expect that the individual contributions would add up to the total, and is the residual given in Table 2 thus an indicator of non-linearity?

2) The authors state towards the end of the introduction section that the "second purpose of this study is to investigate the usefulness of EMIC for long-term simulations of the ocean carbon cycle by comparing our results to previous studies." This sounds like "EMIC" would be a well defined class of models with homogeneous properties, which is not the case. Some of the cited EMICs (e.g. Zickfeld et al. 2008) employ a 3d state-of-the art ocean model, which is not fundamentally different from the ocean model used in this study. The authors do not discuss sufficiently why specific feedbacks could be expected to be present in their model but not in a simpler model. They also do not provide an in depth comparison of their results with previous EMIC studies (which I

would expect for an issue that is the "second purpose of this study"). I actually do not belive that the question as to the "usefulness of EMIC for long-term simulations of the ocean carbon cycle" could be answered in this study - this would require a dedicated model intercomparison study with a common experimental design. I would recommend to drop this "second purpose", and discuss results compared to previous EMIC studies as necessary to place the present study in the scientific context.

Further, the conclusions regarding the "usefulness of EMIC" are inconsistent. On page 9, lines1-2, it is stated that "results support the usefulness of EMIC for long-term projections of the ocean carbon cycle". Later in the "Summary and Discussion" it is speculated about why the simpler models used in previous studies would have significantly different feedback mechanisms. Should this be interpreted as "simpler models are right for the wrong reason, but this is still useful"?

Minor points

page 1, line 14: at year 2000 -> after 2000 model years

page 1, line 22: "...circulation change becomes a second order process." This is in contradiction to the statement that "changes in the biological pump via ocean circulation" is the dominant process.

page 2, line 4: "...over a 1000-year period" -> "on millennial time scales" or similar

page 2, lines 16-18: "In those previous studies...". This assertion is not correct. E.g. Maier-Reimer et al. 1996 state that both biological and physical carbon-climate feedbacks are small compared to the carbon concentration feedback. I do not think that the other cited studies make the point that biology is a second order process (but I have not checked in-depth).

page 4, line 11: setting -> settling (?)

page 4, line 22: "As for spin-up,..." -> "For the spin-up..."

page 5, line 9: an -> the

page 5, lines 22-29: It should be made clearer here which effect is included in which experiment. E.g., the authors state that the experiments GW_om and GW_ca "evaluate the contributions of changes in the organic matter and CaCO3 cycles." This is not very precise, since these experiments evaluate changes in one part of the "cycles" only (changes in production and remineralisation rates, but the rate of upward transport of reminerelised OM is not included).

page 5, line 30: "...are included in not..." check grammar

page 5, line 31: I guess the pre-industrial sea ice fractions are used only in the gas exchange calculations? Please clarify.

page 6, line 8: "... are likely to reflect the non-linearity..." Please describe what the experiments reflect. There is no need to speculate ("likely").

page 7, line 11: "According to..." -> "Consistent with the..."

page 7, line 18: "...rain ratio increasing from 0.09 to 1.13..." Please check the numbers.

page 7, lines 16-17: Please explain briefly why PP increases and export decreases. It is not obvious from the model description why this could happen (if necessary or helpful, please amend the model description accordingly)

page 8, line 5: "of the same simulation using models..." -> "of the corresponding simulations"

page 12, line 33: Plattner et al. 2001 do have abiotic experiments, but they do not use this to quantify the contribution of biology

Figure 3: a separation into panels for surface and deep ocean would be useful (or a stretch of the depth scale in the upper 1000m)

---

## Author Comment (AC1) · 26 Jan 2018

**Response to Reviewer 1 (N.C. Swart)**

General comments This paper seeks to understand the source of carbon climate feedbacks arising in the ocean on multi-centennial timesscales. This is an important question in the Earth System Modelling community, including for understanding future climate change, and interpreting carbon budgets. The authors use a well thought out experimental design to quantify the sensitivity of different aspects of the ocean carbon cycle (e.g. biology, circulation, solubility etc) to climate change. The approach is based on previous work, but fairly novel in this particular application. The paper is well orga-

nized and written, and the results, including the graphics are clear. Most uncertainties are addressed and the results are placed in the context of previous work. I thoroughly enjoyed this paper. Almost every time I had a question it was answered in the follow sentence or section. Overall I assess the quality as very high, and I recommend publication. I don't have any major issues. I do have some comments which I think could help to clarify the paper and address the few lingering questions that I did have.

**Response: We appreciate the positive evaluation and many thoughtful comments from the reviewer. Referring to the comments, we will carefully revise the manuscript. Specific replies are as follows.**

Specific comments o The authors describe a decreasing ocean CO2 uptake under global warming, and attribute this in large part to a reduction in export production. However previous literature (e.g. de Vries et al. [2012], Marinov et al. [2008] and references therein) has shown that ocean CO2 uptake is not directly tied to export production (as one might guess), but rather to the so called "efficiency of the biological pump". Please clarify how export production, biological pump efficiency and carbon uptake relate in this study. Specifically, is it really export production which is important - and if so why is this different from the above literature?

**Response: Thank you for a useful suggestion. In our simulation, globally averaged preformed PO4 increases from 1.15 mmol/m3 in the pre-industrial condition to 1.40 mmol/m3 at the end of the simulation. Export production decreases from 8.1 PgC/yr to 6.3 PgC/yr. Considering the previous literatures pointed out by reviewer, the reduction of oceanic CO2 uptake due to global warming would be attribute to decrease in biological pump efficiency rather than EP reduction in our simulations. We will add the description of relationship between export production, biological pump efficiency and carbon uptake in the revised manuscript. We will also describe the importance of reduction in biological pump efficiency to decreasing CO2 uptake under global warming in the abstract and conclusion.**

[Figure]

o The authors force the offline ocean biogeochemical model with monthly mean fields from the AOGCM (including for insolation, velocity, temperature, salinity etc). This means that much variability is being averaged over, including the diurnal cycle, synoptic scale variability and so on. There is a known sensitivity of ocean model response to forcing frequency. Obviously forcing at a higher frequency means more data, and is more expensive. But please discuss how the results might be sensitive to the forcing frequency. I don't necessarily need any more experiments, just a clear caveat on this point.

**Response: As mentioned below and manuscript, we compared passive salinity tracer in the offline model to online salinity in the AOGCM. There were no significant differences in the salinity distribution between two simulations. Therefore, the short-term processes have limited impact on our results. We will add the discussion in the revised manuscript.**

o Circulation plays a small direct role, but a large indirect role through nutrient transport. The circulation changes are large (and mostly consistent with expectations). In various parts of the manuscript, the authors do a good job of comparing their results to those from CMIP and other studies. If possible it would be interesting to know how the MIROC simulated circulation changes under 4xCO2 compare to other CMIP models. More generally a comment on how sensitive the results are to uncertainties, for example in the climate model response to increasing CO2, would be helpful. (I note the authors do discuss the need for similar studies using different models, but the reasons for this could be fleshed out).

**Response: According to the reviewer's comment, we compare our results to other AOGCMs and EMICs under high CO2 scenario (e.g. 4xCO2 and RCP8.5). The weakening of AMOC and AABW formation in the first 140 years of our simulation are consistent with the results of CMIP5 models under RCP8.5 (Weaver et al., 2012; Heuzé et al., 2015). However, the longer-term responses of AMOC and AABW formation are very uncertain. In our simulation, AMOC shutdown**

**continues to the end of the simulation without recovery. Partial or full AMOC recovery to the pre-industrial level has emerged in other long-term AOGCM and EMIC simulations [Schmittner et al., 2008, Weaver et al., 2012, Li et al., 2013]. AABW formation recovers and overshoots after 1000 years in our simulations. These responses have not been reported in previous multi-millennium simulations [Schmittner et al., 2008, Li et al., 2013]. The uncertainties of circulation changes would have impacts on millennial-scale CO2 uptake. We will add the description of uncertainty of circulation change and its impact on long-term carbon cycle to the discussion in the revised manuscript.**

Technical comments and typos (by pg and ln) pg 1 / Abstract: ln 8: "accelerate an increase in CO2" - Is there really an "acceleration". I'm not sure that this is the right word. I think just "decrease oceanic carbon uptake and therefore increase atmospheric CO2 and global warming" would sound better and be more accurate.

**Response: According to the reviewer's comment, we will modify this sentence in the revised manuscript.**

ln 14: "...first 140 years (at year 2000)" - the meaning of this because clear later when reading the methods, but this could be a little confusing in the abstract, because readers do not know at that point what experiment you are conducting. For example, on first reading I was thinking "calendar year 2000".

**Response: We agree the reviewer's comment. Following the comment of reviewer 3, we will change from "at year 2000" to "after 2000 model years" in the revised manuscript.**

ln 19: "...gradient of DIC substantially" - add a comma after "DIC"

**Response: We will correct this.**

ln 23-4: "uptake through natural carbon cycle" - suggest removing "natural carbon cycle". I don't think this is needed.

**Response: We will remove this in the revised manuscript.**

pg 2: ln 5-6: "...long-term evolution of climate systems with slow response times..." -> "...long term evolution of climate system components with a slow response time..." (i.e. there is only one climate system, which is made up of many components).

**Response: We will revise this sentence following the reviewer's comment.**

ln 10: "accelerating the rate of $CO_2$ accumulation" - again I'm not sure if "accelerating" is accurate? Maybe just "increasing $CO_2$ accumulation in the atmosphere".

**Response: We agree the reviewer's comment. We will modify this sentence in the revised manuscript.**

ln 13: "primarily alter"...delete "primarily". There are only the natural and anthropogenic $CO_2$ cycles.

**Response: We will remove "primarily" in the revised manuscript.**

ln 15-16: Another good study to reference is Randerson et al. (2015). They show that ocean carbon feedbacks become larger than land carbon feedbacks, but only on very long time scales. There is a nice tie in with this work.

**Response: Thank you for the nice suggestion. We will add Randerson et al. (2015) to the reference in the revised manuscript.**

ln 15-20: I suggest mentioning here that you will explain later why those studies came to that conclusion (and are different from yours).

**Response: We will add these information in this paragraph.**

ln 25 "However the contributions"...suggest deleting "However". This sentence is not really a continuation of the previous sentence.

**Response: According to the reviewer's comment, we will remove "However".**

ln 26-27: There are no studies doing this breakdown for CMIP5?

**Response: We do not know this kind of study using CMIP5. We will add "To our knowledge," to this sentence in the revised manuscript.**

ln 28 "with AOGCM" -> "with an AOGCM"

pg 3: ln 3 "using AOGCM" -> "an AOGCM".

ln 13 "with MIROC 4m AOGCM" -> "with the MIROC 4m AOGCM"

**Response: Thank you for pointing out. We will correct these in the revised manuscript.**

ln 27-28 "according to AOGCM climate simulations" - I got what you meant, but this could be clearer. Maybe something like "following the physical evolution of AOGCM climate simulations", or "forced by output from AOGCM climate simulations".

**Response: We will change the sentence to the latter one in the revised manuscript.**

pg 4: ln 11: "setting flux" -> "settling flux"

**Response: Thank you for pointing out. We will fix typo in the revised manuscript.**

ln 16-18: "we confirmed..." - I found this confusing. At the bottom of page 3, it says that salinity is specified from the AOGCM simulations - but here you are saying you are using salinity from the offline simulation to validate against the AOGCM simulation. Something is missing. Do you simulate a passive salinity tracer in the offline model, to compare against the "online" salinity in the AOGCM? Please clarify.

**Response: As reviewer said, we compared passive salinity tracer in the offline model to online salinity in the AOGCM. We will add this description to the revised manuscript.**

ln 25-31: Just noting that the comparison is between a pre-industrial simulation and modern observations. This could have some impact. Are you using GLODAP estimated PI DIC and ALK to compare against? Not a big deal but worth clarifying.

**Response: We compared a pre-industrial simulation with modern observations. We will add this information to revised manuscript for clarifying.**

pg 5: ln 5-7 "This model does not include..." - it seems like these sentences belonged in section 2.2 to me. They are about the model, not the experiment.

**Response: We will move this sentence to section 2.2 following the reviewer's comment.**

ln 9 "We conducted additional experiments"...these were only run for 500 years, right? Maybe worth mentioning here.

**Response: The reviewer is right. We will add this information in the revised manuscript.**

ln 9-20: It is mentioned briefly below, but I think it is worth mentioning clearly here at the outset that the experimental design assumes linearity of the feedbacks.

**Response: We agree the reviewer's comment. We will add the assumption of linearity of the feedbacks to this paragraph in the revised manuscript.**

ln 23: "and oceanic interior temperature and salinity". When I thought about the experimental design - as far as I can tell these interior T and S values are not used for anything in the offline model for this particular experiment, since the organic matter cycle is specified. The SST is, I believe, still be specified as GW. If this is all true, I would just remove the mention of "interior T and S values", since it is not relevant, and could be confusing. If these values are used for something, please clarify.

**Response: This is our mistake. Interior T and S are not used in the sensitivity experiments. We will just remove the mention of "interior T and S values" in the revised manuscript.**

pg 6: ln 12 :"after the summary of the global mean" - a bit confusing as written. Maybe

"...and ocean biogeochemical variables. A full summary of the global mean changes is reported in..."

**Response: According to the reviewer's comment, we will modify this sentence.**

pg 7: ln 2 / fig 1 e: I suggest you add the line for wind stress at year 2000 to Fig 1e (most other panels in fig. 1 are showing a year 2000 result). It would be helpful to see the recovery.

**Response: We agree the reviewer's comment. We will add the line for year 2000 in the revised manuscript.**

ln 6-15: $PO_4$ is shown, but what about $NO_3$? More generally, the paper discusses export production in general, but does not mention how diazotrophs and "other" phytoplankton react?

**Response: Global $NO_3$ at the surface also decreases by about 20%. Regional $NO_3$ changes are similar to the $PO_4$ changes. Diazotrophs and "other" phytoplankton increase slightly, which is consistent with previous study (Schmittner et al., 2008). Increase in "other" phytoplankton is caused by faster nutrient recycling due to seawater warming. We will add these description to the results in the revised manuscript.**

pg 8: ln 6: "...during constant atmospheric CO2..." - I would include the year 140, as in "...constant atmospheric CO2 after year 140..." for clarity.

**Response: We will correct this sentence.**

ln 27-33: I was interested in this section, and would like to see more spatial information. If possible, it would be really nice to see a Hovmoller, like Fig 1a, but for CO2 uptake/flux anomaly (of GW - CTL) (maybe in the SI).

**Response: We agree the reviewer's comment. We will add a Hovmoller figure in the supplementary information.**

pg 9: ln 23-24: I suggest you reference these "uptake change" numbers back to table 2.

**Response: We will add the reference to table 2 in the revised manuscript.**

pg 10: ln 19-23: Le Quere et al 2008 claim that the westerly wind increase is reducing Southern Ocean $CO_2$ uptake (i.e. the opposite of what is being said here). Therefore, it is strange to cite as evidence without further explanation. I suggest it would be better to reference the Zickfeld et al. response to Quere et al. (who show that the $CO_2$ uptake response to wind changes is time-scale dependent). The effect of circulation change on sDIC (Fig 5) is essentially a redistribution of carbon from the Atlantic to the Pacific. Interestingly, we saw a similar redistribution due to to wind stress induced circulation changes in Swart et al. (2012), which we linked back to changes in the Agulhas leakage and overturning circulation.

**Response: Following the reviewer's comment, we will delete the reference to Le Quere et al 2008. We will also add the description of carbon redistribution due to wind stress induced circulation changes to the revised manuscript.**

Figures: 1. e : please add line for year 2000

**Response: We will add the line for year 2000 in the revised manuscript as mentioned above.**

3. The colorbar is not perceptually uniform, which makes it hard to determine where large changes have actually occurred. Please consider using a perceptually uniform colorbar.

**Response: We will change the color bar in the revised manuscript.**

6. Caption "Global upper-ocean" - fix typo

**Response: Thank you for pointing out. We will fix typo.**

**References: Heuzé, C., Heywood, K. J., Stevens, D. P., and Ridley, J. K.: Changes**

**in global ocean bottom properties and volume transports in CMIP5 models under climate change scenarios, J. Climate, 28, 2917-2944, doi:10.1175/JCLI-D-14-00381.1, 2015 Weaver, A. J. SedláÉĞcek, J., Eby, M., Alexander, K., Crespin, E., Fichefet, T., Philippon-Berthier, G., Joos, F., Kawamiya, M., Matsumoto, K., Steinacher, M., Tachiiri, K., Tokos, K., Yoshimori, M., and Zickfeld, K.: Stability of the Atlantic meridional overturning circulation: a model intercomparison, Geophys. Res. Lett., 39, L20709, doi:10.1029/2012GL053763, 2012.**

---

## Author Comment (AC2) · 26 Jan 2018

**Response to Reviewer 2**

The manuscript by Yamamoto et al explores through a large suite of experiments under fixed atmospheric concentrations the role physical changes in climate play on ocean carbon uptake. Their conclusions suggest, in contrast to other papers, that the change of circulation dominate the response. It took me a little while to get into this paper, but once there I enjoyed the paper much and really appreciate the larger number of simulations that went into this work - thank you. Overall this is well conceived and executed piece of work, that will be of interested to a wide readership. I do have some

minor comments that I feel once addressed would strengthen the paper, otherwise I am happy to recommend this paper for publication.

**Response: We appreciate the positive evaluation and helpful comments from the reviewer. Referring to the comments, we will carefully revise the manuscript. Specific replies are as follows.**

Minor Comments: 1. The authors predicate the study on global warming, and state that global warming will decrease ocean carbon uptake. However in the present day, as CO2 levels continue to rise - the ocean will take up carbon at a rate proportional to this i.e. gradient driven. I do understand in this study, if we assume fixed CO2 levels then this supposition is correct, but I do think this needs to clarified in the text.

**Response: The reviewer is right. We will refer to the assumption of fixed CO2 level in the revised manuscript.**

2. The study puts more heat and carbon into the ocean over a much shorter period than under CMIP3/5 change changes runs, even the business-as-usual scenario; this of course has implications for where the heat and carbon are stored. As the authors make a number comparison to these climate change runs - could they comment on what the implications of this maybe - perhaps on the timing of events e.g. sinks to sources etc, and whether its a fair comparison?

**Response: As reviewer pointed out, input of heat and carbon into the ocean during the first 140 year of our experimental design are larger than SRES A2 and RCP8.5. In the first 140 year, the response of climate and oceanic carbon cycle would be somewhat different from the RCP8.5 simulations. After year 140, the influence of the initial differences of heat and carbon input on oceanic carbon cycle would weaken since CO2 concentration is similar between 4xCO2 and RCP8.5. Therefore, we think that the differences between 4xCO2 and RCP8.5 have a limited impact on long-term response of climate and carbon cycle to global warming.**

3. The experimental methods section is super critical to this paper, however I needed to read this at least 5 times to be really clear. I recommend that the authors break up the 3rd paragraph to make it more accessible

**Response: We agree the reviewer's comments. We will break up the 3rd paragraph in the revised manuscript.**

4. The study uses offline simulations, which make sense, could the authors comments on whether on or offline makes much difference - given the challenges of capturing short-term processes in the fields needed to run the model. I am sure that they have tested this somewhere, and if not it should be acknowledged.

**Response: The reviewer is right. We compared passive salinity tracer calculated in the offline model to online salinity in the AOGCM. There were no significant differences in salinity distribution between the two simulations. We will add this information to the revised manuscript.**

5. The timescales calculated in the paper are based on a fixed atmospheric concentrations. In the real world i.e. driven by emissions, the ocean carbon uptake would significantly slow as the gradient between the ocean and atmosphere decreases. I think this probably needs to be mentioned in the discussion, as do the implications for timing of changes.

**Response: As reviewer pointed out, our simulation with prescribed CO2 concentrations are idealized. On the other hand, there is an advantage of the simulations with prescribed CO2 concentrations compared to the simulations with prescribed emissions. The simulations with prescribed CO2 concentrations allow for a more rigorous separation of feedback processes since carbon sinks respond to the same atmospheric CO2 concentration in all simulations (Zickfeld et al., 2011). We will mention the difference between emission driven runs and concentration driven runs and usefulness of concentration driven runs in the revised manuscript.**

6. Otherwise some minor typos etc need to be addressed, but I am sure they will be picked in the proofs.

**Response: We will carefully correct typo and errors in the revised manuscript.**

**References: Zickfeld, K., Eby, M., Matthews, H.D., Schmittner, A., and Weaver, A.J.: Nonlinearity of carbon cycle feedbacks, J. Climate, 24, 4255-4275, 2011.**

---

## Author Comment (AC3) · 26 Jan 2018

**Response to Reviewer 3**

The authors study long term ocean carbon cycle feedbacks over a time horizon of 2000 years by using an offline ocean biogeochemistry model driven by climate model output. By using different combinations of output fields from a control simulation (no global warming) and a global warming simulation, they separate the carbon cycle feedback into components originating from SST-changes, circulation changes, changes of the biological pump, and a few others. They find that changes in the biological pump contribute most to the carbon uptake reduction under climate change followed by solubility changes. The authors claim that this finding is "contrary to most previous studies".

The manuscript is clearly within the scope of Biogeosciences. The main conclusions, however, are partly inconsistent and not well enough supported by the results. Also, the manuscript as it stands now, it is not very novel. Many similar studies on ocean carbon-climate feedbacks have been published during the past 20 years, most of them with simpler models. However, the authors do not convincingly make the point as to why significantly different results could be expected because of enhanced model complexity. There are (or potentially are) interesting new aspects in the present study, but the authors do not elaborate these (see below).

**Response: We are grateful for the careful review. The reviewer's comments helped us to improve our manuscript. Referring to the comments, we will carefully revise the manuscript. Specific replies are as follows.**

Major points:
1)The statement that the results are "contrary to most previous studies" is not convincingly supported by the results presented in this manuscript. Since the experimental set-up is different from (most of the) previous studies, it remains unclear what the effect of these differences might be. This is briefly discussed in section 5, following speculations (page 12, lines 20-30) about why models in previous studies possibly gave different results. These speculations are not convincingly supported by the results or the cited literature either. In my opinion it is most likely that differences in the experimental set-up explain much of the differences. The authors follow Zickfeld et al. (2008) in designing their experiments, and use the tendencies of DIC and ALK due to biological production/remineralisation from the CTL-experiment in the GW-experiment (and vice versa) to determine the effect of biology on CO2 uptake. This mimics pre-industrial organic matter and CaCO3 production/remineralisiation under a reduced circulation. I find this design questionable, since it weakens the upward transport of
remineralised carbon and nutrients (leading to enhanced C-uptake), but at the same time keeps the export production at pre-industrial levels (leading also to enhanced uptake). The experiment design used in some of the previous studies (Joss et al. 1999, Plattner et al. 2001) is different: Here, archived pre-industrial surface sPO4 and sALK fields are used in a global warming simulation to separate the "effect of biology". If I am not mistaken, the effect of reduced upwelling of DIC is cancelled out in this experiment. Other studies (Sarmiento et al. 1998, Matsumoto et al. 2010) use abiotic experiments.

**Response: As for the upward transport of remineralized carbon and nutrients, we think that the lack of our explanation of experimental design misled the reviewer. To quantify the effect of biology on CO2 uptake, we compared the GW-experiment to the GW-experiment with biological production/remineralization from the CTL-experiment (GW_om) ((2) - (6) in Table 2). In the GW-experiment, the upward transport of remineralized carbon and nutrients are weakened by both circulation change and the reduction of remineralization. In the GW_om, the upward transport of remineralized carbon and nutrients are weakened by only circulation change since biological production/remineralization are kept at pre-industrial levels. Comparing the GW_om to the GW-experiment, the upward transport of remineralized carbon and nutrients are enhanced, leading to reduced carbon uptake. We will describe experimental details in the section 2.3 "Experimental design".**

In order to demonstrate that the feedback-mechanisms are really substantially different from previous studies, the authors would need to quantify the differences arising due to the different experimental set-up (or different interpretations of the "biological effect"). This could be done by running additional sets of experiments following the design of previous studies. A discussion of which experimental set-up or definition of "biological pump contribution" is more useful or correct should also be

provided. The authors state in the abstract that "quantifications of the contributions from different processes to the overall reduction in ocean uptake are still unclear". Instead of adding to the confusion they could take the opportunity to assess what the experimental set-up in different studies contributes to this. This would also be a novel and useful contribution to the field.

**Response: We agree the reviewer's comments. In fact, before submitting the manuscript, we conducted the abiotic experiments following the previous studies (Sarmiento et al. 1998 and Matsumoto et al 2010) (attached Table 1). In our abiotic experiments, the reduction in oceanic $CO_2$ uptake associated with global warming is caused by changes in the ocean circulation and SST (attached Figure 1). Biological effect slightly enhances $CO_2$ uptake. These contributions of individual mechanisms are consistent with previous estimation using abiotic experiments (Sarmiento et al. 1998 and Matsumoto et al 2010). Therefore, the different results are caused by the differences of experiments set-up, as pointed by reviewer.**

**We attribute these different results to the vertical gradient of DIC under the pre-industrial condition of the abiotic and biotic experiments (i.e., CTL-base/GW-base experiments) (attached Figure 2). The effect of circulation change in the abiotic experiments mainly represents the reduced $CO_2$ uptake due to weaker downward transport of anthropogenic $CO_2$ from the surface to the deep ocean. The enhanced $CO_2$ uptake associated with the reduced upward transport of natural $CO_2$ from the deep ocean to the upper ocean is underestimated. This is because the vertical gradient of DIC in the abiotic experiments is much smaller than the observed DIC gradient. The increase in $CO_2$ uptake due to weaker equatorial upwelling is not found in the abiotic experiments (attached figure 3).**

**In the circulation effect of CTL-base/ GW-base experiments with the realistic vertical gradient of DIC, the enhanced CO2 uptake associated with the reduced upward transport of natural CO2 largely offsets the reduced CO2 uptake due to weaker downward transport of anthropogenic CO2 (as mentioned in section 4.2) . Therefore, the reduction in CO2 uptake due to circulation change is larger in the abiotic experiments than in the CTL-base/GW-base experiments.**

**In the abiotic experiments, the contribution of biological effect is calculated as the residual (attached Table 2). The enhanced CO2 uptake owing to the reduced upward transport of natural CO2 is included in the biological effect. This effect overcomes the reduced CO2 uptake due to the weakening of the biological pump, so that biological effect shows an increase in CO2 uptake. Our results show that CTL-base/GW-base experiments are useful for quantifying the contribution of circulation and biological change.**

**We will add these comparison and discussion in the revised manuscript. Unfortunately, due to computer resources, it is difficult to conduct additional experiments based on other previous studies (Joss et al. 1999, Plattner et al. 2001) immediately. However, we believe the comparison of two different experimental set-up (CTL-base/GW-base and abiotic experiments) is useful for understanding of the feedback mechanisms.**

In this context it would be also useful to discuss the limitations of the approaches to separate the feedback-mechanisms in a non-linear system. The authors have already performed two sets of experiments (GW-base and CTL-base), which could serve this purpose. Results from these sets of experiments are presented in Table 2, but are only mentioned in one brief sentence (page 9, lines 24-25) in the manuscript. Particularly, for the "Biology"-contribution, the authors find a considerable dependence on the base state (GW or CTL; 118 PgC difference while the total is 402 PgC). An explanation for

this would be useful. Do the authors expect that the individual contributions would add up to the total, and is the residual given in Table 2 thus an indicator of non-linearity?

**Response: According to the reviewer's comment, we will explain the dependence of biological effect on the base state in the revised manuscript.**

2) The authors state towards the end of the introduction section that the "second purpose of this study is to investigate the usefulness of EMIC for long-term simulations of the ocean carbon cycle by comparing our results to previous studies." This sounds like "EMIC" would be a well defined class of models with homogeneous properties, which is not the case. Some of the cited EMICs (e.g. Zickfeld et al. 2008) employ a 3d state-of the art ocean model, which is not fundamentally different from the ocean model used in this study. The authors do not discuss sufficiently why specific feedbacks could be expected to be present in their model but not in a simpler model. They also do not provide an in depth comparison of their results with previous EMIC studies (which I would expect for an issue that is the "second purpose of this study"). I actually do not belive that the question as to the "usefulness of EMIC for long-term simulations of the ocean carbon cycle" could be answered in this study - this would require a dedicated model intercomparison study with a common experimental design. I would recommend to drop this "second purpose", and discuss results compared to previous EMIC studies as necessary to place the present study in the scientific context. Further, the conclusions regarding the "usefulness of EMIC" are inconsistent. On page 9, lines1-2, it is stated that "results support the usefulness of EMIC for long-term projections of the ocean carbon cycle". Later in the "Summary and Discussion" it is speculated about why the simpler models used in previous studies would have significantly different feedback mechanisms. Should this be interpreted as "simpler models are right for the wrong reason, but this is still useful"?

**Response: We agree the reviewer's comment. We will remove this "sec-**

ond purpose", the sentence on page 9, lines 1-2, and speculation about different feedback mechanisms of the simpler models in the "Summary and Discussion". We will simply compare our results with previous EMIC studies in the revised manuscript

Minor points
page 1, line 14: at year 2000 -> after 2000 model years

**Response: We will modify this according to the reviewer's comment.**

page 1, line 22: "...circulation change becomes a second order process." This is in contradiction to the statement that "changes in the biological pump via ocean circulation" is the dominant process.

**Response: We agree the reviewer's comments. We will rewrite this sentence to "circulation change plays a small direct role, but a large indirect role through nutrient transport and biological pump." in the revised manuscript.**

page 2, line 4: "...over a 1000-year period" -> "on millennial time scales" or similar

**Response: According to the reviewer's comment, we will modify this.**

page 2, lines 16-18: "In those previous studies...". This assertion is not correct. E.g. Maier-Reimer et al. 1996 state that both biological and physical carbon-climate feedbacks are small compared to the carbon concentration feedback. I do not think that the other cited studies make the point that biology is a second order process (but I have not checked in-depth).

**Response: We will remove the reference of Maier-Reimer et al (1996) and revise this sentence.**

page 4, line 11: setting -> settling (?)

**Response: We will fix typo in the revised manuscript.**

page 4, line 22: "As for spin-up,..." -> "For the spin-up..."

**Response: Following the reviewer's comment we will modify this in the revised manuscript.**

page 5, line 9: an -> the

**Response: We will correct this in the revised manuscript.**

page 5, lines 22-29: It should be made clearer here which effect is included in which experiment. E.g., the authors state that the experiments GW_om and GW_ca "evaluate the contributions of changes in the organic matter and CaCO3 cycles." This is not very precise, since these experiments evaluate changes in one part of the "cycles" only (changes in production and remineralisation rates, but the rate of upward transport of reminerelised OM is not included).

**Response: In the comparison between GW and GW_om, the changes in upward transport of remineralized OM due to reduced OM remineralization are included as mentioned in the response to the major points 1). The changes in upward transport of remineralized OM due to circulation change are calculated in the comparison between GW and GW_circ. According to the reviewer's comment, we will add experimental details to the revised manuscript.**

page 5, line 30: "...are included in not..." check grammar

**Response: Thank you for pointing out. We will correct this.**

page 5, line 31: I guess the pre-industrial sea ice fractions are used only in the gas exchange calculations? Please clarify.

**Response: The reviewer is right. We will clarify this sentence in the revised manuscript.**

page 6, line 8: "... are likely to reflect the non-linearity..." Please describe what the experiments reflect. There is no need to speculate ("likely").

**Response: We will remove "likely" in the revised manuscript.**

page 7, line 11: "According to..." -> "Consistent with the..."

**Response: Following the reviewer's comment we will correct this.**

page 7, line 18: "...rain ratio increasing from 0.09 to 1.13..." Please check the numbers.

**Response: Thank you for pointing out. We will change from 1.13 to 0.13 in the revised manuscript.**

page 7, lines 16-17: Please explain briefly why PP increases and export decreases. It is not obvious from the model description why this could happen (if necessary or helpful, please amend the model description accordingly)

**Response: PP increase are also found in Schmittner et al (2008) and Taucher and Oschlies (2011). Our model is based on Schmittner et al (2008). Taucher and Oschlies (2011) show that PP increase is caused by temperature effects on biological processes such as remineralization and the microbial loop. We will add these information in the revised manuscript.**

page 8, line 5: "of the same simulation using models..." -> "of the corresponding simulations"

**Response: Thank you for pointing out. We will correct this.**

page 12, line 33: Plattner et al. 2001 do have abiotic experiments, but they do not use this to quantify the contribution of biology

**Response: We will delete this reference in the revised manuscript.**

Figure 3: a separation into panels for surface and deep ocean would be useful (or a stretch of the depth scale in the upper 1000m)

**Response: According to the reviewer's comment, we will revise Figure 3.**

**Reference:**
**Taucher, J., and Oschlies, A.: Can we predict the direction of marine primary production change under global warming?, Geophys. Res. Lett., 38, L02603, doi:10.1029/2010GL045934, 2011.**

**Figure caption**
**Figure 1. Contributions of the mechanisms to the reduction in the oceanic CO2**

uptake due to global warming at 500 years for the abiotic experiments. The total change and the contribution of the individual mechanisms are calculated as summarized in Table 2. We will add this figure to the Figure 4 in the manuscript.

Figure 2.   Vertical profile of the difference in salinity-normalized DIC from the surface under the pre-industrial condition. The black and red lines show the abiotic and biotic models, respectively.

Figure 3.   Zonal mean change in the salinity-normalized DIC induced by circulation changes for the abiotic experiments at 500 years. The left and right panels show the Atlantic and Indo-Pacific Oceans, respectively. We will add this figure to the Figure 5 in the manuscript.

Table 1.   Description of the abiotic experiments and results of the oceanic CO2 uptake. We will add this table to the Table 1 in the manuscript.

Table 2.   The contributions of individual mechanisms to the reduction in the CO2 uptake due to global warming in the first 500 years under the abiotic experiments. We will add this table to the Table 2 in the manuscript.

[Figure]

**Fig. 1.**

[Figure]

Fig. 2.

**BGD**

Atlantic Ocean (Abiotic)

Indian-Pacific Ocean (Abiotic)

(mmol/m³)

-200  -160  -120  -80  -40  0  40  80  120  160  200

sDIC change by circulation change

**Fig. 3.**

| | Experiments | Changing mechanisms | Cumulative uptake (Pg C) | | SST | Dilution | Circulation | Organic matter cycle | CaCO$_3$ cycle |
|---|---|---|---|---|---|---|---|---|---|
| | | | 500year | 2000year | | | | | |
| 1 | CTL | – | 1629 | 2888 | CTL | CTL | CTL | CTL | CTL |
| 2 | GW | all | 1227 | 2028 | GW | GW | GW | GW | GW |
| 3 | CTL_abio | – | 1819 | | CTL | CTL | CTL | – | – |
| 4 | GW_abio | SST circulation | 1371 | | GW | GW | GW | – | – |
| 5 | GW_abio_SST | SST | 1511 | | CTL | GW | GW | – | – |

**Fig. 4.** Table 1

| Mechanisms | | uptake change (Pg C) |
|---|---|---|
| Total | $(2) - (1)$ | -402 |
| SST | $(4) - (5)$ | -140 |
| Freshwater | $-$ | $-$ |
| Circulation | $(5) - (3)$ | -308 |
| Biology | $(\text{Total}) - (\text{SST}) - (\text{Biology})$ | 46 |

**Fig. 5.** Table 2

---

## Author Response (AR1)

Dear Editor

We are grateful to you and the three reviewers for the helpful comments on the previous version of our manuscript. We have addressed all the comments made by the three reviewers, as follows. Generally, we revised the manuscript and conducted additional experiments (abiotic experiments) according to reviewer's comments. This is because the comments of all reviewers are very useful for improving our manuscript and strengthening the interpretation of our model results. Native speaker has performed proofreading of our manuscript and corrected errors and inappropriate expression in English sentences We hope that the revised version of our paper is now suitable for publication in Biogeosciences.

Response to Reviewer #1 (N.C. Swart)
(Our response highlighted gray.)

General comments

This paper seeks to understand the source of carbon climate feedbacks arising in the ocean on multi-centennial timesscales. This is an important question in the Earth System Modelling community, including for understanding future climate change, and interpreting carbon budgets. The authors use a well thought out experimental design to quantify the sensitivity of different aspects of the ocean carbon cycle (e.g. biology, circulation, solubility etc) to climate change. The approach is based on previous work, but fairly novel in this particular application. The paper is well organized and written, and the results, including the graphics are clear. Most uncertainties are addressed and the results are placed in the context of previous work. I thoroughly enjoyed this paper. Almost every time I had a question it was answered in the follow sentence or section. Overall I assess the quality as very high, and I recommend publication. I don't have any major issues. I do have some comments which I think could help to clarify the paper and address the few lingering questions that I did have.

Response: We appreciate the positive evaluation and many thoughtful comments from the reviewer. Referring to the comments, we will carefully revise the manuscript. Specific replies are as follows.

Specific comments

• The authors describe a decreasing ocean CO2 uptake under global warming, and attribute this in large part to a reduction in export production. However previous literature (e.g. de Vries et al. [2012], Marinov et al. [2008] and references therein) has shown that ocean CO2 uptake is not directly tied to export production (as one might guess), but rather to the so called "efficiency of the biological pump". Please clarify how export production, biological pump efficiency and carbon uptake relate in this study. Specifically, is it really export production which is important - and if so why is this different from the above literature?

Response: Thank you for a useful suggestion. In our simulation, globally averaged preformed $PO_4$ increases from 1.15 mmol/m$^3$ in the pre-industrial condition to 1.40 mmol/m$^3$ at the end of the simulation. Export production decreases from 8.1 PgC/yr to 6.3 PgC/yr. Considering the previous literatures pointed out by reviewer, the reduction of oceanic $CO_2$ uptake due to global warming would be attribute to decrease in biological pump efficiency rather than EP reduction in our simulations. We added the changes in biological pump efficiency to page 9, lines 7-11 in the revised manuscript (page 22, lines 7-11 in this response). We also described the importance of reduction in biological pump efficiency to decreasing $CO_2$ uptake under global warming in the abstract and conclusion.

• The authors force the offline ocean biogeochemical model with monthly mean fields from the AOGCM (including for insolation, velocity, temperature, salinity etc). This means that much variability is being averaged over, including the diurnal cycle, synoptic scale variability and so on. There is a known sensitivity of ocean model response to forcing frequency. Obviously forcing at a higher frequency means more data, and is more expensive. But please discuss how the results might be sensitive to the forcing frequency. I don't necessarily need any more experiments, just a clear caveat on this point.

Response: As mentioned below and manuscript, we compared passive salinity tracer in the offline model to online salinity in the AOGCM. There were no significant differences in the salinity distribution between two simulations. Therefore, the short-term processes have limited impact on our results. We added this information to page 5, lines 1-5 in the revised manuscript (page 18, lines 1-5 in this response).

• Circulation plays a small direct role, but a large indirect role through nutrient transport. The circulation changes are large (and mostly consistent with expectations). In various parts of the manuscript, the authors do a good job of comparing their results to those from CMIP and other studies. If possible it would be interesting to know how the MIROC simulated circulation changes under $4xCO_2$ compare to other CMIP models. More generally a comment on how sensitive the results are to uncertainties, for example in the climate model response to increasing $CO_2$, would be helpful. (I note the authors do discuss the need for similar studies using different models, but the reasons for this could be fleshed out).

Response: According to the reviewer's comment, we compare our results to other AOGCMs and EMICs under high $CO_2$ scenario (e.g. $4xCO_2$ and RCP8.5). The weakening of AMOC and AABW formation in the first 140 years of our simulation are consistent with the results of CMIP5 models under RCP8.5 (Weaver et al., 2012; Heuzé et al., 2015). However, the longer-term responses of AMOC and AABW formation are very uncertain. In our simulation, AMOC shutdown continues to the end of the simulation without recovery. Partial or full AMOC recovery to the pre-industrial level has emerged in other long-term AOGCM and EMIC simulations [Schmittner et al., 2008, Weaver et al., 2012, Li et al., 2013]. AABW formation recovers and overshoots after 1000 years in our simulations. These responses have not been reported in previous multi-millennium simulations [Schmittner et al., 2008, Li et al., 2013]. The uncertainties of circulation changes would have impacts on millennial-scale $CO_2$ uptake. We added the description of uncertainty of circulation change and its impact on long-term carbon cycle to page 16, lines 1-10 in the revised manuscript (page 29, lines 1-10 in this response).

Technical comments and typos (by pg and ln)
pg 1 / Abstract:
ln 8: "accelerate an increase in CO2" - Is there really an "acceleration". I'm not sure that this is the right word. I think just "decrease oceanic carbon uptake and therefore increase atmospheric CO2 and global warming" would sound better and be more accurate.

Response: According to the reviewer's comment, we modified this sentence in page 1, lines 8-9 in the revised manuscript (page 14, lines 8-9 in this response).

ln 14: "...first 140 years (at year 2000)" - the meaning of this because clear later when reading the methods, but this could be a little confusing in the abstract, because readers do not know at that point what experiment you are conducting. For example, on first reading I was thinking "calendar year 2000".

Response: We agree the reviewer's comment. Following the comment of reviewer #3, we changed from "at year 2000" to "after 2000 model years" in page 1, lines 15 in the revised manuscript (page 14, lines 15 in this response).

ln 19: "...gradient of DIC substantially" - add a comma after "DIC"

Response: We added "and" instead of comma in page 1, line 19 (page 14, line 19 in this response).

ln 23-4: "uptake through natural carbon cycle" - suggest removing "natural carbon cycle". I don't think this is needed.

Response: We removed this in the revised manuscript (page 14, lines 24, right column in this response).

pg 2:
ln 5-6: "...long-term evolution of climate systems with slow response times..." -> "...long term evolution of climate system components with a slow response time..." (i.e. there is only one climate system, which is made up of many components).

Response: We revised this sentence following the reviewer's comment in page 2, lines 5-6 (page 15, lines 5-6 in this response).

ln 10: "accelerating the rate of CO2 accumulation" - again I'm not sure if "accelerating" is accurate? Maybe just "increasing CO2 accumulation in the atmosphere".

Response: We agree the reviewer's comment. We modified this sentence in page 2, lines 10-11 (page 15, lines 10-11 in this response).

ln 13: "primarily alter"...delete "primarily". There are only the natural and anthropogenic CO2 cycles.

Response: We removed "primarily" in the revised manuscript (page 15, line 13, right column in this response).

ln 15-16: Another good study to reference is Randerson et al. (2015). They show that ocean carbon feedbacks become larger than land carbon feedbacks, but only on very long time scales. There is a nice tie in with this work.

Response: Thank you for the nice suggestion. We added Randerson et al. (2015) to the reference in page 2, line 12 (page 15, line 12 in this response).

ln 15-20: I suggest mentioning here that you will explain later why those studies came to that conclusion (and are different from yours).

Response: We added these information to page 3, lines 7-16 (page 16, lines 7-16 in this response).

ln 25 "However the contributions"...suggest deleting "However". This sentence is not really a continuation of the previous sentence.

Response: We removed "However" in the revised manuscript (page 15, line 30, right column in this response).

ln 26-27: There are no studies doing this breakdown for CMIP5?

Response: We do not know this kind of study using CMIP5. We added "To our knowledge," to this sentence in page 2, line 30 (page 15, line 30 in this response).

ln 28 "with AOGCM" -> "with an AOGCM"

pg 3:
ln 3 "using AOGCM" -> "an AOGCM".

ln 13 "with MIROC 4m AOGCM" -> "with the MIROC 4m AOGCM"

Response: Thank you for pointing out. We corrected these in page 3, line 1; page 3, line 4; page 3 line 25 (page 16, line 1; page 16, line 4; page 16 line 25in this response).

ln 27-28 "according to AOGCM climate simulations" - I got what you meant, but this could be clearer. Maybe something like "following the physical evolution of AOGCM climate simulations", or "forced by output from AOGCM climate simulations".

Response: We changed the sentence to the latter one in page 4, lines 7-8 (page 17, lines 7-8 in this response).

pg 4:
ln 11: "setting flux" -> "settling flux"

Response: Thank you for pointing out. We fixed typo in page 4, line 25 (page 17, line 25 in this response).

ln 16-18: "we confirmed..." - I found this confusing. At the bottom of page 3, it says that salinity is specified from the AOGCM simulations - but here you are saying you are using salinity from the offline simulation to validate against the AOGCM simulation. Something is missing. Do you simulate a passive salinity tracer in the offline model, to compare against the "online" salinity in the AOGCM? Please clarify.

Response: As reviewer said, we compared passive salinity tracer in the offline model to online salinity in the AOGCM. We added this description to page 5 lines1-5 (page 18, lines 1-5 in this response).

ln 25-31: Just noting that the comparison is between a pre-industrial simulation and modern observations. This could have some impact. Are you using GLODAP estimated PI DIC and ALK to compare against? Not a big deal but worth clarifying.

Response: We compared a pre-industrial simulation with modern observations. We added this information to page 5 lines 15-16 (page 18, lines 15-16 in this response).

pg 5:
ln 5-7 "This model does not include..." - it seems like these sentences belonged in section 2.2 to me. They are about the model, not the experiment.

Response: We moved this sentence to section 2.2 (page 4, lines 29-31) (page 17, lines 29-31 in this response).

ln 9 "We conducted additional experiments"...these were only run for 500 years, right? Maybe worth mentioning here.

Response: The reviewer is right. We added this information to page 6, line 4 (page 19, line 4 in this response).

ln 9-20: It is mentioned briefly below, but I think it is worth mentioning clearly here at the outset that the experimental design assumes linearity of the feedbacks.

Response: We agree the reviewer's comment. We added the assumption of linearity of the feedbacks to page 6 line 11 (page 19, line 11 in this response).

ln 23: "and oceanic interior temperature and salinity". When I thought about the experimental design - as far as I can tell these interior T and S values are not used for anything in the offline model for this particular experiment, since the organic matter cycle is specified. The SST is, I believe, still be specified as GW. If this is all true, I would just remove the mention of "interior T and S values", since it is not relevant, and could be confusing. If these values are used for something, please clarify.

Response: This is our mistake. Interior T and S are not used in the sensitivity experiments. We just remove the mention of "interior T and S values" in the revised manuscript (page 20, line 5, right column in this response).

pg 6:
ln 12 :"after the summary of the global mean" - a bit confusing as written. Maybe "...and ocean biogeochemical variables. A full summary of the global mean changes is reported in..."

Response: We modified this sentence in page 8, lines 3-4 (page 21, lines 3-4 in this response).

pg 7:
ln 2 / fig 1 e: I suggest you add the line for wind stress at year 2000 to Fig 1e (most other panels in fig. 1 are showing a year 2000 result). It would be helpful to see the recovery.

Response: We agree the reviewer's comment. We added the line for year 2000 in the revised manuscript (Fig. 2e).

ln 6-15: PO4 is shown, but what about NO3? More generally, the paper discusses export production in general, but does not mention how diazotrophs and "other" phytoplankton react?

Response: Global $NO_3$ at the surface also decreases by about 20%. Regional $NO_3$ changes are similar to the $PO_4$ changes. Diazotrophs and "other" phytoplankton increase slightly, which is consistent with previous study (Schmittner et al., 2008). Increase in "other" phytoplankton is caused by faster nutrient recycling due to seawater warming. We added these description to page 8, line 30 and page 9, line 14 (page 21, line 30 and page 22, line 14 in this response).

pg 8:
ln 6: "...during constant atmospheric CO2..." - I would include the year 140, as in "...constant atmospheric CO2 after year 140..." for clarity.

Response: We corrected this sentence in page 10, line 7 (page 23, line 7 in this response).

ln 27-33: I was interested in this section, and would like to see more spatial information. If possible, it would be
5 really nice to see a Hovmoller, like Fig 1a, but for CO2 uptake/flux anomaly (of GW - CTL) (maybe in the SI).

Response: We agree the reviewer's comment. We added a Hovmoller figure to the supplementary figure 4.

pg 9:
10 ln 23-24: I suggest you reference these "uptake change" numbers back to table 2.

Response: We added the reference to table 2 to page 12 line 7 (page 25, line 7 in this response).

pg 10:
15 ln 19-23: Le Quere et al 2008 claim that the westerly wind increase is reducing Southern Ocean CO2 uptake (i.e.
the opposite of what is being said here). Therefore, it is strange to cite as evidence without further explanation. I
suggest it would be better to reference the Zickfeld et al. response to Quere et al. (who show that the CO2 uptake
response to wind changes is time-scale dependent). The effect of circulation change on sDIC (Fig 5) is essentially
a redistribution of carbon from the Atlantic to the Pacific. Interestingly, we saw a similar redistribution due to to
20 wind stress induced circulation changes in Swart et al. (2012), which we linked back to changes in the Agulhas
leakage and overturning circulation.

Response: Following the reviewer's comment, we will delete the reference to Le Quere et al 2008. We also added
the description of carbon redistribution due to wind stress induced circulation changes to page 11 lines 25-27
25 (page 24, lines 25-27 in this response).

Figures:
1. e : please add line for year 2000

30 Response: We added the line for year 2000 in the revised manuscript (Fig. 2e) as mentioned above.

3. The colorbar is not perceptually uniform, which makes it hard to determine where large changes have actually
occurred. Please consider using a perceptually uniform colorbar.

35 Response: We changed the color bar in the revised manuscript (Fig. 4).

6. Caption "Global upper-ocean" - fix typo

Response: Thank you for pointing out. We fixed typo in page 28, line 2 (page 41, line 2 in this response).
40

Response to Reviewer #3
(Our response highlighted gray.)

The authors study long term ocean carbon cycle feedbacks over a time horizon of 2000 years by using an offline ocean biogeochemistry model driven by climate model output. By using different combinations of output fields from a control simulation (no global warming) and a global warming simulation, they separate the carbon cycle feedback into components originating from SST-changes, circulation changes, changes of the biological pump, and a few others. They find that changes in the biological pump contribute most to the carbon uptake reduction under climate change followed by solubility changes. The authors claim that this finding is "contrary to most previous studies".

The manuscript is clearly within the scope of Biogeosciences. The main conclusions, however, are partly inconsistent and not well enough supported by the results. Also, the manuscript as it stands now, it is not very novel. Many similar studies on ocean carbon-climate feedbacks have been published during the past 20 years, most of them with simpler models. However, the authors do not convincingly make the point as to why

5 significantly different results could be expected because of enhanced model complexity. There are (or potentially are) interesting new aspects in the present study, but the authors do not elaborate these (see below).

Response: We are grateful for the careful review. The reviewer's comments helped us to improve our manuscript. Referring to the comments, we will carefully revise the manuscript. Specific replies are as follows.

Major points:
1) The statement that the results are "contrary to most previous studies" is not convincingly supported by the results presented in this manuscript. Since the experimental set-up is different from (most of the) previous studies, it remains unclear what the effect of these differences might be. This is briefly discussed in section 5, following

15 speculations (page 12, lines 20-30) about why models in previous studies possibly gave different results. These speculations are not convincingly supported by the results or the cited literature either. In my opinion it is most likely that differences in the experimental set-up explain much of the differences. The authors follow Zickfeld et al. (2008) in designing their experiments, and use the tendencies of DIC and ALK due to biological production/remineralisation from the CTL-experiment in the GW-experiment (and vice versa) to determine the

20 effect of biology on CO2 uptake. This mimics pre-industrial organic matter and CaCO3 production/remineralisiation under a reduced circulation. I find this design questionable, since it weakens the upward transport of remineralised carbon and nutrients (leading to enhanced C-uptake), but at the same time keeps the export production at pre-industrial levels (leading also to enhanced uptake). The experiment design used in some of the previous studies (Joss et al. 1999, Plattner et al. 2001) is different: Here, archived pre-

25 industrial surface sPO4 and sALK fields are used in a global warming simulation to separate the "effect of biology". If I am not mistaken, the effect of reduced upwelling of DIC is cancelled out in this experiment. Other studies (Sarmiento et al. 1998, Matsumoto et al. 2010) use abiotic experiments.

Response: As for the upward transport of remineralized carbon and nutrients, we think that the lack of our

30 explanation of experimental design misled the reviewer. To quantify the effect of biology on $CO_2$ uptake, we compared the GW-experiment (GW_bio) to the GW-experiment with biological production/remineralization from the CTL-experiment (GW_bio_om) ((2) – (9) in Table 2). In the GW_bio, the upward transport of remineralized carbon and nutrients are weakened by both circulation change and the reduction of remineralization. In the GW_bio_om, the upward transport of remineralized carbon and nutrients are weakened by only circulation

35 change since biological production/remineralization are kept at pre-industrial levels. Comparing the GW_bio_om to the GW_bio, the upward transport of remineralized carbon and nutrients are enhanced, leading to reduced carbon uptake. We added the experimental details to Page 7, lines 11-13 in the revised manuscript (page 20, lines 11-13 in this response).

40 In order to demonstrate that the feedback-mechanisms are really substantially different from previous studies, the authors would need to quantify the differences arising due to the different experimental set-up (or different interpretations of the "biological effect"). This could be done by running additional sets of experiments following the design of previous studies. A discussion of which experimental set-up or definition of "biological pump contribution" is more useful or correct should also be provided. The authors state in the abstract that

45 "quantifications of the contributions from different processes to the overall reduction in ocean uptake are still

unclear". Instead of adding to the confusion they could take the opportunity to assess what the experimental set-up in different studies contributes to this. This would also be a novel and useful contribution to the field.

Response: We agree the reviewer's comments. In fact, before submitting the original manuscript, we conducted the abiotic experiments following the previous studies (Sarmiento et al. 1998 and Matsumoto et al 2010) (attached Table 1 and 2). In our abiotic experiments, the reduction in oceanic $CO_2$ uptake associated with global warming is caused by changes in the ocean circulation and SST (attached Figure 1). Biological effect slightly enhances $CO_2$ uptake. These contributions of individual mechanisms are consistent with previous estimation using abiotic experiments (Sarmiento et al. 1998 and Matsumoto et al 2010). Therefore, the different results are caused by the differences of experiments set-up, as pointed by reviewer.

We added the methods of abiotic experiment and it's results to Section 2.4.1 (page 6), Section 4.1 (page 11), Fig. 1, Fig. 4, Fig. 5, Table 1 and Table 2. We also discuss the reason for the different estimations between the abiotic and CTL-base/GW-base experiments in Section 4.3 (page 14-15).

Unfortunately, due to computer resources, it is difficult to conduct additional experiments based on other previous studies (Joss et al. 1999, Plattner et al. 2001) immediately. However, we believe the comparison of two different experimental set-up (CTL-base/GW-base and abiotic experiments) is useful for understanding of the feedback mechanisms.

In this context it would be also useful to discuss the limitations of the approaches to separate the feedback-mechanisms in a non-linear system. The authors have already performed two sets of experiments (GW-base and CTL-base), which could serve this purpose. Results from these sets of experiments are presented in Table 2, but are only mentioned in one brief sentence (page 9, lines 24-25) in the manuscript. Particularly, for the "Biology"-contribution, the authors find a considerable dependence on the base state (GW or CTL; 118 PgC difference while the total is 402 PgC). An explanation for this would be useful. Do the authors expect that the individual contributions would add up to the total, and is the residual given in Table 2 thus an indicator of non-linearity?

Response: According to the reviewer's comment, we added the explanation of non-linear response of biological effect in CTL-base/GW-base experiments to Page 14, lines 4-10 (page 27, lines 4-10 in this response).

2) The authors state towards the end of the introduction section that the "second purpose of this study is to investigate the usefulness of EMIC for long-term simulations of the ocean carbon cycle by comparing our results to previous studies." This sounds like "EMIC" would be a well defined class of models with homogeneous properties, which is not the case. Some of the cited EMICs (e.g. Zickfeld et al 2008) employ a 3d state-of the art ocean model, which is not fundamentally different from the ocean model used in this study. The authors do not discuss sufficiently why specific feedbacks could be expected to be present in their model but not in a simpler model. They also do not provide an in depth comparison of their results with previous EMIC studies (which I would expect for an issue that is the "second purpose of this study"). I actually do not belive that the question as to the "usefulness of EMIC for long-term simulations of the ocean carbon cycle" could be answered in this study - this would require a dedicated model intercomparison study with a common experimental design. I would recommend to drop this "second purpose", and discuss results compared to previous EMIC studies as necessary to place the present study in the scientific context. Further, the conclusions regarding the "usefulness of EMIC" are inconsistent. On page 9, lines1-2, it is stated that "results support the usefulness of EMIC for long-term projections of the ocean carbon cycle". Later in the "Summary and Discussion" it is speculated about why the

simpler models used in previous studies would have significantly different feedback mechanisms. Should this be interpreted as "simpler models are right for the wrong reason, but this is still useful"?

Response: We agree the reviewer's comment. We removed this "second purpose", and speculation about different feedback mechanisms in the simpler models in the "Summary and Discussion". We simply compared our results with previous EMIC studies in the revised manuscript.

Minor points

page 1, line 14: at year 2000 -> after 2000 model years

Response: We modified this in page 1, lines 15 (page 14, line 15 in this response).

page 1, line 22: "...circulation change becomes a second order process." This is in contradiction to the statement that "changes in the biological pump via ocean circulation" is the dominant process.

Response: We rewrote this sentence to "circulation change plays a small direct role, but a large indirect role through nutrient transport and biological processes." in Page 1, lines 23-24 (page 14, lines 23-24 in this response).

page 2, line 4: "...over a 1000-year period" -> "on millennial time scales" or similar

Response: We modified this in page 2, line 4 (page 15, line 4 in this response).

page 2, lines 16-18: "In those previous studies...". This assertion is not correct. E.g. Maier-Reimer et al. 1996 state that both biological and physical carbon-climate feedbacks are small compared to the carbon concentration feedback. I do not think that the other cited studies make the point that biology is a second order process (but I have not checked in-depth).

Response: We removed the reference in page 2, lines 14-16 (page 15, lines 14-16 in this response).

page 4, line 11: setting -> settling (?)

Response: We fixed typo in page 4, line 25 (page 17, line 25 in this response).

page 4, line 22: "As for spin-up,..." -> "For the spin-up..."

Response: Following the reviewer's comment we modified this in page 5, line 9 (page 18, line 9 in this response).

page 5, line 9: an -> the

Response: We corrected this in page 6, line 4 (page 19, line 4 in this response).

page 5, lines 22-29: It should be made clearer here which effect is included in which experiment. E.g., the authors state that the experiments GW_om and GW_ca "evaluate the contributions of changes in the organic matter and CaCO3 cycles." This is not very precise, since these experiments evaluate changes in one part of the "cycles"

only (changes in production and remineralisation rates, but the rate of upward transport of reminerelised OM is not included).

Response: In the comparison between GW_bio and GW_bio_om, the changes in upward transport of remineralized OM due to reduced OM remineralization are included as mentioned in the response to the major points 1). The changes in upward transport of remineralized OM due to circulation change are calculated in the comparison between GW_bio and GW_bio_circ. According to the reviewer's comment, we added experimental details to page 7, lines 11-13 (page 20, lines 11-13 in this response).

page 5, line 30: "...are included in not..." check grammar

Response: Thank you for pointing out. We corrected this in page 7, line 20 (page 20, line 20 in this response).

page 5, line 31: I guess the pre-industrial sea ice fractions are used only in the gas exchange calculations? Please clarify.

Response: The reviewer is right. We clarified this sentence in page 7, lines 18-20 (page 20, lines 18-20 in this response).

page 6, line 8: "... are likely to reflect the non-linearity..." Please describe what the experiments reflect. There is no need to speculate ("likely").

Response: We removed "likely" (page 20, line 30, right column in this response).

page 7, line 11: "According to..." -> "Consistent with the..."

Response: We corrected this in page 9, lines 1-2 (page 22, lines 1-2 in this response).

page 7, line 18: "...rain ratio increasing from 0.09 to 1.13..." Please check the numbers.

Response: We changed from 1.13 to 0.13 in page 9, line 20 (page 22, line 20 in this response).

page 7, lines 16-17: Please explain briefly why PP increases and export decreases. It is not obvious from the model description why this could happen (if necessary or helpful, please amend the model description accordingly)

Response: PP increase are also found in Schmittner et al (2008) and Taucher and Oschlies (2011). Our model is based on Schmittner et al (2008). Taucher and Oschlies (2011) show that PP increase is caused by temperature effects on biological processes such as remineralization and the microbial loop. We added these information to page 9, lines 13-16 (page 22, lines 13-16 in this response).

page 8, line 5: "of the same simulation using models..." -> "of the corresponding simulations"

Response: We corrected this in page 10, line 5 (page 23, lines 5 in this response).

page 12, line 33: Plattner et al. 2001 do have abiotic experiments, but they do not use this to quantify the contribution of biology

Response: We deleted Plattner et al. 2001 in the reference of abiotic experiments.

Figure 3: a separation into panels for surface and deep ocean would be useful (or a stretch of the depth scale in the upper 1000m)

Response: According to the reviewer's comment, we revised Figure 3 (Figure 4 in the revised manuscript).

Reference:

[revised manuscript text omitted]

*Pre-industrial seawater temperature is used for calculating biologi

**Seawater temperature of GW_bio is applied for calculating biolo

[revised manuscript text omitted]

---

## Author Response (AR2)

Response to Reviewer

(Our response highlighted gray.)

I have previously reviewed a first version of this manuscript. I think the authors have done a fairly good job in revising their manuscript. I have, however, two important points and several minor/technical issues that should be addressed.

Response: We appreciate many thoughtful comments from the reviewer. According to the comments, we carefully revised the manuscript. Specific replies are as follows.

Main points:

1) In Section 2.4.1, which describes the abiotic experiments, we read "The biological effect is calculated as the residual ("Total" - "SST" - "Circulation", as described in Table 2)." Then, in Section 4.1 (Abiotic experiments): "Biological changes enhance $CO_2$ uptake at high latitudes and reduce it at low latitudes; the resulting global $CO_2$ uptake increase by 46 PgC due to biological changes." Later, on page 14 (lines 29 to 30): "In the abiotic experiments, the biological contribution is calculated as the residual. The enhanced $CO_2$ uptake owing to the reduced upward transport of natural $CO_2$ is included in the biological effect." Also, in Figure 5 panel a, which displays the results from the abiotic experiments there are grey bars indicating an "organic matter contribution".

What kind of abiotic experiment did the authors perform here? This needs clarification.

Response: Thank you for pointing out. As written in Section 2.4.1, we conducted abiotic experiment based on the previous studies (Sarmiento et al., 1998; Matsumoto et al., 2010). In the abiotic experiments, the biological effect is calculated as the residual based on Sarmiento et al (1998) (see the caption of Table 1 in Sarmiento et al 1998). We think that description of Section 4.1 and Figure 5 in the previous manuscript confused reviewer. We added "Biological change is calculated as the residual" to Section 4.1 (Page 16, line 2 in this response). In Figure 5, we changed "organic matter contribution" to "biological contribution". We also added "biological contribution is calculated as the residual" to the caption (Page 30, lines 4-5 in this response).

2) On page 9 lines 6-11, and later scattered throughout the text: It is rather surprising that the efficiency of the biological pump (calculated in terms of AOU) is going down. This is the opposite of what is seen in CMIP5 models for the 1% simulation (Schwinger et al. 2014). In all of these models (no exception) the lower export production is overcompensated by the slower circulation such that AOU increases. So in all of these models the efficiency of the biological pump is actually going UP not DOWN. Could the authors try to explain why this is different for their model (or at least mention that this is

different to CMIP5 models)? Since the reduced efficiency is used as an explanation for changes in section 4.2.3, I think this is important.

Response: Thank you for useful comment. In our model, $P_{remi}/P_{tot}$ increases from its preindustrial value of 0.47 to 0.51 in the first 200 years. This increased efficiency of the biological pump is consistent with CMIP5 models. After 500 model years, $P_{remi}/P_{tot}$ decrease to 0.34 by the end of the simulation. Preformed $PO_4$ is accumulated in AABW (attached figure and supplementary figure 3). This reduced efficiency of the biological pump is caused by enhanced AABW formation after 500 model years (Yamamoto et al., 2015). We added these descriptions to the revised manuscript (Page 13, lines 10-14 in this response). We also rewrote the first paragraph of section 4.2.3 (Page 17, lines 20-28 in this response).

Minor points and technical corrections:

page 4, lines 17-18: "...a physiological trade-off between the efficiency of nutrient encounters at the cell surface and the maximum assimilation rate...". Should this read "encountered"?

Response: Thank you for pointing out. We changed "encounters" to "encountered" (Page 8, line 17 in this response).

page 4, lines 29-31: Using "although" does not make sense here, please check logic of this scentence.

Response: We checked and rewrote this sentence (Page 8, lines 29-30 in this response).

page 5, line 2: "climate variability" is not the correct term here. This should read "short-term variability" or "forcing variability" or something similar.

Response: We changed "climate variability" to "short-term variability" according to your suggestion (Page 9, lines 1-2 in this response).

page 6, line 11: "...assuming that the individual contributions to total change are linear." I do not understand this addition. The nonlinearity is quantified in Table 2, and briefly discussed for the biological pump contribution in Sec. 4.2.3. So I don't see where an assumption is made.

Response: We agreed reviewer's comment and remove this sentence (Page 10, line 11 in this response).

page 7, line 15-16: Consider replacing "effects" with "sensitivity experiments" or "experiments" or similar.

Response: We replaced "effects" with "sensitivity experiments" according to reviewer's comment (Page 11, line 16 in this response).

page 11, line 10: "the previous" -> "previous"

Response: Thank you for pointing out. We will correct this (Page 15, line 13 in this response).

page 11, line 14, and Table 2: The total for the abiotic experiments should be calculated as (4)-(3) I assume, not (2)-(1) as done here. It would be very surprising if the total uptake for the abiotic experiments is the same as in the biotic.

Response: We think that this comment is reviewer's misunderstanding. In previous studies (Sarmiento et al., 1998; Matsumoto et al., 2010), the total uptake was calculated using biotic experiments. Abiotic experiments are used only for separating the physical effect (SST and circulation effects) from total uptake change. According to previous studies, total uptake is calculated as (2) – (1) in this study.

page 12, lines 8-9: I think it is better to write "... a significant nonlinearity ... is found for the organic matter cycle."

Response: We wrote this sentence according to reviewer's comment (Page 16, line 12 in this response).

page 12, line 9: It would help the reader to add a sentence "The large differences relative to the abiotic simulation are discussed below (Sec. 4.3)." or something similar.

Response: We added this sentence according to reviewer's comment (Page 16, lines 13-14 in this response).

page 12, line 18: "...secondary importance of circulation changes". I would suggest to replace the "secondary importance" as in other places of the revised manuscript, since this is misleading with respect to the role of circulation changes via biology.

Response: We agree reviewer's comment. W remove "secondary importance" (Page 16, line 24 in this response).

page 15, lines 14-15: "...with the previous studies" -> "...with those previous studies"

Response: Thank you for pointing out. We corrected this (Page 19, line 26 in this response) .

page 15, line 19: "...under the pre-industrial condition" -> "...under pre-industrial conditions"

Response: We corrected this (Page 19, line 31 in this response).

Figure 1: The caption says this is salinity-normalised DIC. But in the Figure title it reads "Delta sDIC", while the the range (-100 to 300 mmol/m^3) indicates that this must be some change or deviation. It is however not clear from what. Please clarify.

[revised manuscript text omitted]